# FuseLIP:
# Multimodal Embeddings via Early Fusion of Discrete Tokens

## Abstract

Contrastive language-image pre-training aligns features of text-image pairs in a common latent space via distinct encoders for each modality. While this approach achieves impressive performance in several zero-shot tasks, it cannot natively handle multimodal inputs, i.e., encoding image and text into a single feature vector. As a remedy, it is common practice to use additional modules to merge the features extracted by unimodal encoders. In this work, we present FuseLIP, a new architecture for multimodal embedding. Leveraging recent progress in discrete image tokenizers, we propose to use a single transformer model operating on a unified vocabulary of text and image tokens. This early fusion approach allows the different modalities to interact at each depth of encoding and obtain richer representations compared to common late fusion. We collect new datasets for multimodal pre-training and evaluation, designing challenging tasks for multimodal encoders. We show that FuseLIP outperforms late fusion approaches in several multimodal and unimodal embedding tasks.

## 1 Introduction

Contrastive language-image pre-training (CLIP) is a fundamental approach for learning semantically rich text and image representations (Radford et al., 2021). The resulting text and image encoders perform well in many zero-shot tasks and have been successfully applied to image generation (Ramesh et al., 2022), transfer learning (Wortsman et al., 2022; Chen et al., 2024), and multimodal large language models (Liu et al., 2023a; Beyer et al., 2024). To improve CLIP-like models, various refinements of encoder architectures, training data, and optimization schemes have been proposed (Cherti et al., 2023; Zhai et al., 2023). However, these models are not designed to extract representations from *multimodal inputs*, i.e., encoding an image-text pair into a single feature vector, as text and images are processed by two separate encoders. Several techniques have adapted pre-trained CLIP models for multimodal retrieval (Liu et al., 2023b; Wei et al., 2024; Zhang et al., 2024) or other downstream tasks (Singh et al., 2022). These methods typically merge features extracted by frozen unimodal encoders through either fixed functions or learnable modules. A different line of work trains multimodal sequence-to-sequence models (Wang et al., 2022; Lu et al., 2022; Mizrahi et al., 2023) using an encoder-decoder architecture (Raffel et al., 2020). While well-suited for transfer learning, these models lack the strong zero-shot capabilities of CLIP.

In this work, we propose a novel multimodal embedding method that extends CLIP to multimodal inputs while preserving its strong vision-language alignment and zero-shot capabilities. Our first key novelty is to encode all input modalities (image, text, and their combinations) using a *single encoder*. We achieve this by leveraging a discrete image tokenizer (Yu et al., 2024), which together with the standard text tokenizer maps inputs into unified sequences of tokens drawn from a *finite multimodal vocabulary*. Since the image tokenizer is trained solely for image compression and reconstruction, it does not introduce any bias with respect to text–image alignment. Processing tokenized inputs with a single encoder, i.e., *early fusion*, allows modalities to interact at every encoding stage, which differs from *late fusion* where the deeper representations from unimodal encoders are merged (Liu et al., 2023b; Zhang et al., 2024). Notably, we can train our architecture, named FuseLIP, with a contrastive loss similar to standard CLIP despite using a single encoder. Moreover, discrete tokenizers enable us to seamlessly incorporate a masked multimodal modeling (MMM) loss into our training objective without the need for multiple auxiliary modules or the computational overhead incurred

by prior work (Singh et al., 2022). Combining the MMM loss with the contrastive objective consistently enhances FuseLIP's performance across various zero-shot tasks (classification, retrieval, VQA, grounding), surpassing or matching late fusion baselines.

To comprehensively evaluate multimodal embedding models, we introduce novel tasks and datasets designed to test modality interactions. We show that late fusion approaches struggle to solve tasks where visual information of the image is more relevant than semantic content, e.g. recognizing a correctly oriented crop. Conversely, our early fusion architecture does not show this limitation, as both modalities communicate at every level of encoding after tokenization. Finally, we demonstrate that training with *hard negative* examples is essential for successfully learning these multimodal tasks. Our work uncovers several interesting aspects of design, training, and evaluation of multimodal embedding models, and may impact future research in this area.

**Contributions.** In summary, our work

- introduces FuseLIP, a novel multimodal embedding model, based on early fusion of discrete image and text tokens, processed by a single transformer encoder, which achieves performance surpassing or comparable to existing late fusion methods,
- shows that FuseLIP can be effectively trained on both unimodal and multimodal data using a contrastive loss and incorporating hard negative examples, while natively supporting and significantly benefiting from a masked modeling objective,
- proposes novel evaluation tasks for multimodal embedding models, complementary to existing benchmarks, highlighting the importance of early fusion and hard negative examples.

## 2 Related Work

**Unimodal embedding.** Popular methods for vision-language pre-training such as CLIP (Radford et al., 2021), SigLIP (Zhai et al., 2023) and ALIGN (Jia et al., 2021) use separate networks to embed each modality. An image encoder and a text encoder, with disjoint parameters, map data into a shared space. Image-text pairs with corresponding semantics are aligned via a contrastive loss on large image-caption datasets. These models achieve good zero-shot performance in tasks like image classification and retrieval, where inputs to be encoded are from a single modality.

**Multimodal embedding.** Certain tasks require encoding multimodal inputs into a single feature vector, which typical methods like CLIP cannot directly handle. Alternative approaches merge representations from text and image encoders. In *score-level fusion* (Liu et al., 2023b), unimodal embeddings are simply summed (possibly with weighting). This method avoids introducing additional merging modules. In contrast, *feature-level fusion* (Singh et al., 2022; Zhang et al., 2024) feeds unimodal embeddings to an additional network, typically a transformer architecture with multiple attention layers. Moreover, BridgeTower (Xu et al., 2023) interconnects features within the latter blocks of the vision (CLIP) and language encoders (RoBERTa): the resulting model is fine-tuned on various downstream tasks. Caffagni et al. (2025) employ a recurrent cell that integrates textual and visual features from intermediate layers of a pre-trained CLIP model. Finally, VisualBERT (Li et al., 2019) and UNITER (Chen et al., 2020) merge visual representations from pre-trained convolutional networks with text tokens and process these with a transformer, training for individual downstream tasks explicitly. In contrast, we use a discrete image tokenizer and focus on zero-shot evaluations.

**Conversion of large multimodal models.** Recent works (Jiang et al., 2024; 2025; Gu et al., 2025; Xue et al., 2025) convert autoregressive large multimodal models (LMMs) into encoders. Such conversion is achieved by fine-tuning an LMM with contrastive learning to return semantically aligned feature vectors. This strategy leverages the large pre-training datasets of the LMM to obtain rich representations, but comes with high inference costs due to their large size.

**Multimodal embedding for composed image retrieval.** Several embedding methods have been proposed for multimodal and composed image retrieval. One approach uses adapter modules (Saito et al., 2023; Baldrati et al., 2023; Gu et al., 2024) or fine-tuning (Zhou et al., 2024) of a pre-trained vision encoder to map its output to a text encoder's input space, enabling image-text encoding. Notably, these approaches rely on pre-trained CLIP (and BERT) models, and are specialized for retrieval tasks.

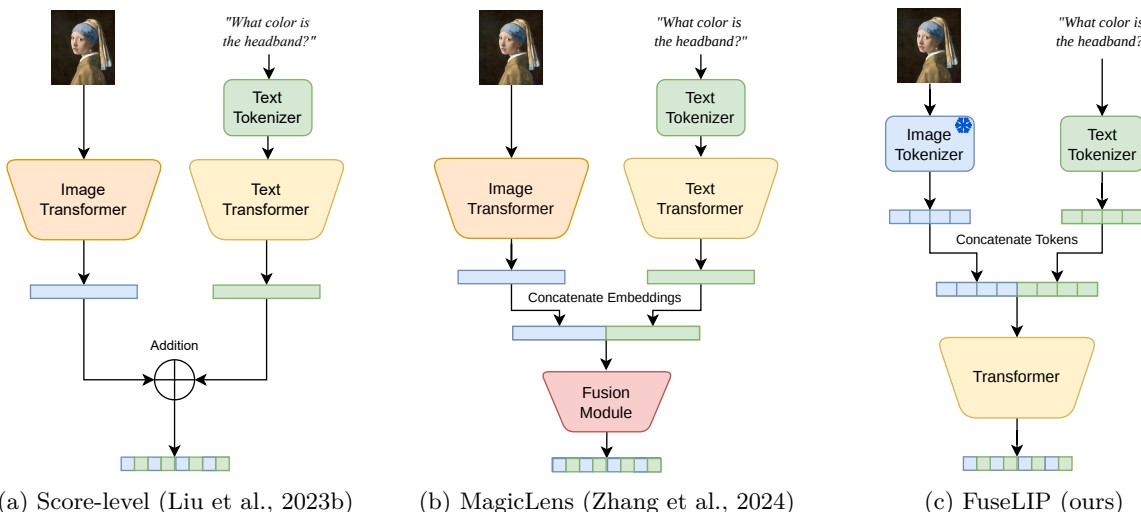

(a) Score-level (Liu et al., 2023b)    (b) MagicLens (Zhang et al., 2024)    (c) FuseLIP (ours)

Figure 1: **Comparison of architectures.** To obtain a multimodal embedding via contrastive learning, late fusion approaches first extract unimodal representations via unimodal encoders, then merge by addition (Liu et al., 2023b) or a fusion module (Zhang et al., 2024). Conversely, our FuseLIP uses a frozen image tokenizer to tokenize inputs of any modality into tokens from a unified vocabulary, which are then processed by a single encoder model. This approach leads to a simple architecture and early fusion of modalities.

**Early fusion in masked multimodal modeling.** Mizrahi et al. (2023) and Bachmann et al. (2024) train encoder-decoder architectures with masked modeling loss on multimodal data, using tokenized and pixel-based RGB images for vision datasets. These models can generate data across modalities and adapt to different tasks via full fine-tuning, but lack zero-shot capabilities enabled by CLIP's contrastive pre-training.

**Early fusion in other domains.** Early fusion has also been explored in other domains. Namely for image generation by DALL·E (Ramesh et al., 2021), and for autoregressive models by Chameleon (Meta, 2024). In contrast, our work focusses on embedding models and the direct comparison to late fusion baselines.

## 3 FuseLIP: Single-Encoder Multimodal Embedding via Early Fusion

### 3.1 Architecture

Late fusion approaches merge image and text representations only in later layers, after independent encoding. Thus, each modality has limited influence on the final features since the modalities communicate only when already heavily processed. However, a multimodal embedding should be strongly conditioned on both input modalities. For instance, in Fig. 1, the joint embedding of the image and the query *"What color is the headband?"* should align with that of *"Blue"*, and exclude irrelevant details. To achieve this, we propose *(a)* enabling early interaction between modalities by merging them immediately after tokenization and *(b)* processing them with a single encoder.

**Tokenization.** While text tokenization is straightforward, compressing images into tokens is more complex. We leverage recent progress in discrete image tokenizers (Van Den Oord et al., 2017; Esser et al., 2021), and use those from the TiTok family (Yu et al., 2024) (frozen during our training), encoding each image into 128 discrete tokens. This ensures symmetry between image and text tokenization, allowing the transformer-based encoder to operate over a finite vocabulary. This property allows us to use a masked multimodal modeling (MMM) loss, without ad-hoc tokenizers or extra computation, unlike FLAVA (Singh et al., 2022). The MMM loss significantly improves model performance (see Sec. 5.3). Moreover, TiTok tokenizers are trained for image reconstruction on ImageNet without text-guided semantic alignment and thereby avoid bias towards a specific representation that may arise during pre-training. In contrast, VISTA (Zhou et al., 2024) initializes

its vision encoder from CLIP, inheriting biases from contrastive pre-training. Finally, TiTok tokenizers provide high-quality compression, as shown by their excellent reconstruction properties (Yu et al., 2024).

**Early fusion.** Text and image are tokenized separately with a single vocabulary of non-overlapping text and vision tokens, obtained by extending the standard text vocabulary with the image token vocabulary. The resulting token sequences are concatenated (images first, followed by text), with special beginning and end of text tokens (`<bot>`, `<eot>`) to separate modalities. Unimodal inputs omit the missing modality (an empty string is appended to image-only inputs). The tokens are then mapped to $d$-dimensional vectors by an embedding matrix and combined with an additive positional embedding that is applied jointly to the full concatenated sequence. Finally, the sequence is processed by the unified transformer encoder.

**Encoder.** We adopt the transformer-based architecture of the SigLIP text encoder, consisting of transformer blocks with bidirectional attention, thereby yielding a bidirectional encoder. An attention mask is applied to exclude empty tokens from the self-attention computation. We use the final-layer representation of the `<eot>` token as the output embedding, which aggregates information from all non-padding tokens through bidirectional attention. Overall, our model can be represented as $f_{\boldsymbol{\theta}_{\text{tok}}, \boldsymbol{\theta}_{\text{enc}}} : I \times T \rightarrow \mathbb{R}^d$, mapping a multimodal input $(\boldsymbol{i}, \boldsymbol{t})$ (which may be unimodal when one modality is missing) to a $d$-dimensional feature vector. It is parameterized by the weights of the tokenizer $\boldsymbol{\theta}_{\text{tok}}$ and encoder $\boldsymbol{\theta}_{\text{enc}}$.

**Auxiliary prediction head.** For the masked modeling loss, we introduce a classification head to predict masked tokens. This module maps the output of the final transformer block to predictions over the token vocabulary $V$ and follows the FLAVA architecture (Singh et al., 2022): a two-layer network with shared embedding and unembedding matrices. We denote this head as $h_{\boldsymbol{\theta}_{\text{head}}} : \mathbb{R}^d \rightarrow \mathbb{R}^{|V|}$, parameterized by $\boldsymbol{\theta}_{\text{head}}$.

## 3.2 Training objective

**Contrastive loss.** To match the zero-shot performance of popular language-image pre-training methods, we optimize the sigmoid loss from SigLIP (Zhai et al., 2023). For a standard dual encoder with separate visual and text towers, and a batch $B$ of image-text pairs $\{(\boldsymbol{i}_k, \boldsymbol{t}_k)\}_{k=1}^{|B|}$, the SigLIP loss is

$$\mathcal{L}_{\text{SigLIP}} = \frac{1}{|B|} \sum_{r=1}^{|B|} \sum_{s=1}^{|B|} \log \left(1 + e^{z_{rs}(-t\phi(\boldsymbol{i}_r) \cdot \psi(\boldsymbol{t}_s) + b)}\right),$$

where $\phi(\boldsymbol{i}_r)$ and $\psi(\boldsymbol{t}_s)$ are normalized embeddings from the vision and text encoders, $z_{rs} = 1$ for positive pairs and $-1$ otherwise, and $t, b$ are learnable parameters. For FuseLIP, the single multimodal encoder $f$ processes multimodal pairs $\{(\boldsymbol{z}_k^1, \boldsymbol{z}_k^2)\}_{k=1}^{|B|}$ into normalized embeddings, where $\boldsymbol{z}_k^1$ and $\boldsymbol{z}_k^2$ can be text, image, or image-text inputs. The loss is then rewritten as

$$\mathcal{L}_{\text{SigLIP}}^{\text{MM}} = \frac{1}{|B|} \sum_{r=1}^{|B|} \sum_{s=1}^{|B|} \log \left(1 + e^{z_{rs}\left(-tf(\boldsymbol{z}_r^1) \cdot f(\boldsymbol{z}_s^2) + b\right)}\right).$$

**Masked modeling loss.** Masked modeling is a popular pre-training approach for both text (Devlin et al., 2019) and image tasks (Bao et al., 2022), where one aims to recover input tokens that have been masked. FLAVA (Singh et al., 2022) shows its effectiveness in multimodal embeddings. However, FLAVA relies on late fusion, using separate vision and text encoders merged by a multimodal module similar to a vision transformer (Dosovitskiy et al., 2021). Since the vision, text and multimodal embedding are provided by different branches of its architecture, a head for each input modality type is added to predict the masked tokens. Because its vision encoder produces continuous embeddings, an additional discrete tokenizer is required for masked modeling, increasing both computational cost and parameter count.

The FuseLIP architecture simplifies the application of masked modeling loss to train multimodal encoders. Since all input modalities are mapped to discrete tokens and processed by the same encoder, additional tokenizers and multiple prediction heads are not necessary. Moreover, the masked modeling and contrastive losses are applied *on the same masked input*, avoiding extra computational overhead. As shown in Sec. 5.3, this strategy significantly improves performance across tasks. In practice, each token (except special tokens)

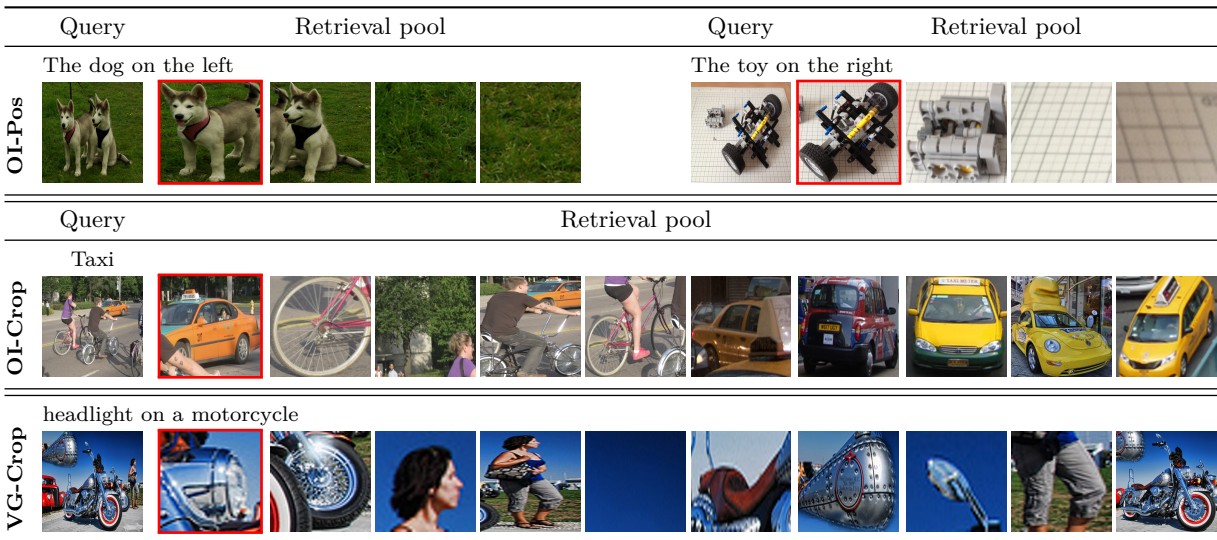

Figure 2: **OI-Pos, OI-Crop and VG-Crop tasks.** We show examples of these tasks (described in Sec. 5.1). The retrieval pool of OI-Crop comprises crops from the same image, as well as the target object cropped from other images. For OI-Pos and VG-Crop it contains only crops of the query image. We show the whole retrieval pool for OI-Pos, OI-Crop, and a sample for VG-Crop. Red frames highlight the ground-truths.

is replaced with a `<MASK>` token with probability $p = 0.1$ (this is distinct from the attention masking applied to empty input tokens). Denoting $J(\boldsymbol{z})$ as the set of masked positions of the masked tokens for an input $\boldsymbol{z}$, $Y(\boldsymbol{z})$ as the corresponding labels, and $f^l$ as the output of the last transformer block, the masked multimodal modeling (MMM) loss for the batch $B = \{(\boldsymbol{z}_k^1, \boldsymbol{z}_k^2)\}_{k=1}^{|B|}$ is

$$\mathcal{L}_{\text{MMM}} = \frac{1}{|B|} \sum_{r=1}^{|B|} \sum_{i=1}^{2} \sum_{(j,y) \in (J(\boldsymbol{z}_r^i), Y(\boldsymbol{z}_r^i))} \mathcal{L}_{\text{CE}} \left( h(f_j^l(\boldsymbol{z}_r^i)), y \right),$$

with cross entropy loss $\mathcal{L}_{\text{CE}}$ and prediction head $h$.

**Final objective.** The final optimization problem combines the two losses: $\min_{\boldsymbol{\theta}_{\text{enc}}, \boldsymbol{\theta}_{\text{head}}} \mathcal{L}_{\text{SigLIP}}^{\text{MM}} + \alpha \, \mathcal{L}_{\text{MMM}}$, where $\alpha$ balances the two losses ($\alpha = 0.25$ in all experiments). The image tokenizer remains frozen during training, meaning that $\boldsymbol{\theta}_{\text{tok}}$ is not optimized. Both objectives are non-causal, therefore FuseLIP remains a bidirectional encoder.

## 4    Training Data

### 4.1    Unimodal data

To train FuseLIP and baseline models, we collect a variety of unimodal and multimodal data. We refer to image-text (I – T) pairs data as unimodal since they do not require joint encoding of inputs from different modalities. These datasets are commonly used to train CLIP-like models, and can also be leveraged by multimodal encoders. We use CC3M (Sharma et al., 2018) and CC12M (Changpinyo et al., 2021), as they provide high-quality images and captions, and are amenable to training within academic compute constraints.

### 4.2    Multimodal data

Datasets with multimodal inputs are scarce. Thus, besides relying on existing ones, we generate additional multimodal data from unimodal data, effectively scaling dataset size at minimal additional collection cost.

**Text-guided image transformations (TGIT).** We generate multimodal data from image datasets by applying transformations and describing them in text. Specifically, we form pairs $(\boldsymbol{i} \oplus \boldsymbol{t}, \boldsymbol{i}')$, where $\boldsymbol{i}$ is the

original image, $\boldsymbol{t}$ is the transformation description in natural language, and $\boldsymbol{i}'$ is the transformed image. We consider transformations such as *random cropping*, *random rotations*, *flipping*, *colorization*, and *color jittering*. See App. A.1 for a detailed description and examples. We apply this approach to a subset of CC3M and CC12M but it can be extended to any image dataset. Since text prompts describe only the transformations, without semantic information about the image, the model must rely on both modalities and cannot solve the task using only text or image.

**VQA data generated from image-text datasets (CC3M-VQA).** We generate VQA data from CC3M by prompting an LLM (Llama-3.1-8B-Instruct from Dubey et al. (2024)) to rewrite captions as question-answer pairs using a structured system prompt (see Fig. 4 in Appendix). We generate 2.4M VQA samples. Notably, this method is scalable to any image-caption dataset. While generating VQA from image-text pairs was explored by Changpinyo et al. (2022), we leverage the recent advancements in LLMs for a simplified and unsupervised generation pipeline.

**VQA data from Visual Genome (VG-VQA).** We use the existing VQA samples from Visual Genome (VG) (Krishna et al., 2017), denoted as VG-VQA, in the standard IT – T format.

**Visual Grounding with Visual Genome (VG-Crop).** VG contains images with rich annotations, in particular natural language descriptions of image regions that are bounded by rectangular boxes. We use these descriptions to build a training dataset, referred to as VG-Crop. Namely, given an image $\boldsymbol{i}$ and a textual description $\boldsymbol{d}$ of some region $\boldsymbol{i}'$ in $\boldsymbol{i}$, we form pairs $(\boldsymbol{i} \oplus \boldsymbol{d}, \boldsymbol{i}')$. That is, the model is tasked to find the crop of the image, given a natural language instruction. Thus, the modality combination is IT – I, which is complementary to that of VQA datasets.

**HQ-Edit.** Finally, HQ-Edit (Hui et al., 2025) consists of synthetically generated image edits. We integrate this dataset into our training by tasking the model to find the correctly edited image. Given an image $\boldsymbol{i}$, an edit described in natural language $\boldsymbol{e}$, and the edited image $\boldsymbol{i}'$, we form pairs $(\boldsymbol{i} \oplus \boldsymbol{e}, \boldsymbol{i}')$. Using the inverse edits $\boldsymbol{e}'$ that are contained in HQ-Edit, we also form the corresponding inverse pairs $(\boldsymbol{i}' \oplus \boldsymbol{e}', \boldsymbol{i})$. As HQ-Edit also contains captions $\boldsymbol{c}'$ of the edited images, we build additional training samples as $(\boldsymbol{i} \oplus \boldsymbol{e}, \boldsymbol{i}' \oplus \boldsymbol{c}')$. This covers both IT – I and IT – IT modalities.

### 4.3 Training with hard negatives

For training, we merge all datasets described in Sections 4.1 and 4.2, ensuring that batches contain diverse tasks and modalities. Due to the contrastive nature of the SigLIP loss, each query is contrasted against all non-matching targets in the batch. Thus, the choice of batch composition determines which negatives the model observes during training. Closely related samples, called *hard negatives*, have been shown to improve contrastive learning (Kalantidis et al., 2020; Robinson et al., 2021; Zhang et al., 2024). We design and integrate hard negatives into the training of FuseLIP and baselines by ensuring that batches contain semantically similar examples. As a result, non-matching cross-combinations between related samples act as hard negatives: the model must, for example, distinguish the correct transformation among several transformations of the same image, or the correct region among several regions from the same image. Specifically, for CC3M-TGIT and CC12M-TGIT we sample multiple transformations of the same image, for VG-Crop and VG-VQA each batch includes three additional samples from the same query image, with different descriptions or questions, and for HQ-Edit we include for every sample the corresponding inverse edit sample in the batch. We show in Sec. 5.3 the key role of hard negatives for learning these tasks.

## 5 Experiments

**Models.** We train two versions of FuseLIP: FuseLIP-S uses the TiTok-S tokenizer and a small transformer as implemented by OpenCLIP (Cherti et al., 2023), while FuseLIP-B uses the TiTok-B tokenizer and a base transformer. We consider two late fusion baselines: *(1)* score fusion (SF), where the multimodal embedding is obtained by summing the unimodal embeddings from text and vision encoders, and *(2)* MagicLens fusion (MLF) (Zhang et al., 2024), which uses a transformer module to merge the unimodal embedding vectors (see Fig. 1). Both baselines use text and vision encoders from CLIP with ViT-S or ViT-B architecture (Dosovitskiy et al., 2021),

and are trained on the SigLIP loss using the same datasets and hard-negatives as FuseLIP. We denote them as SigLIP$_{SF}$ and SigLIP$_{MLF}$ respectively. As we train all models from scratch, we do not compare against methods that fine-tune pre-trained models (Zhang et al., 2024; Jiang et al., 2025). In Table 1, we report the parameter count for all architectures: FuseLIP-S has a total number of parameters similar to the S-sized baselines, but significantly fewer trainable ones as the image tokenizer remains frozen. In contrast, the total parameter amount of FuseLIP-B roughly matches that of B-sized baselines, while the

Table 1: **Parameter comparison** (in million). Greyed numbers indicate non-trainable parameters.

| Model | Image | Text | Fusion | Trained | Total |
|---|---|---|---|---|---|
| SigLIP-S$_{SF}$ | 21.8 | 40.5 | - | 62.3 | 62.3 |
| SigLIP-S$_{MLF}$ | 21.8 | 40.5 | 7.7 | 70.0 | 70.0 |
| SigLIP-B$_{SF}$ | 86.2 | 63.5 | - | 149.7 | 149.7 |
| SigLIP-B$_{MLF}$ | 86.2 | 63.5 | 13.7 | 163.4 | 163.4 |
| FuseLIP-S | 25.9 | - | 42.1 | 42.1 | 68.0 |
| FuseLIP-B | 86.6 | - | 65.6 | 65.6 | 152.2 |

trainable parameter amount of FuseLIP-B is similar to the S-sized baselines. This discrepancy in trainable versus total parameters also affects the training cost: training of FuseLIP is faster and requires less GPU memory as shown in App. A.2.

**Training.** We train 8 epochs (total of 93M seen samples) on CC3M plus multimodal data (CC3M+MM), and on CC12M plus multimodal data (CC12M+MM) we train 16 epochs (326M samples), see Table 2. Full training hyperparameters are provided in App. A.3.

### 5.1 Multimodal evaluation tasks

For evaluation, we consider a variety of embedding tasks. Besides testing on VG-Crop and CC3M-TGIT, used during training, we collect existing and new datasets described here (details in App. A.4). The diversity of these tasks enables a more comprehensive evaluation of different model capabilities. On each task, given a query, the model has to retrieve the correct target from a set of candidate targets, as detailed below. Queries and candidates are embedded and the model selects the the candidate with the highest cosine similarity to the query in the embedding space.

**Massive Multimodal Embedding Benchmark.** MMEB (Jiang et al., 2025) is a multimodal embedding benchmark consisting of 36 subtasks that are split into the categories *Classification*, *VQA*, *Retrieval*, and *Grounding* and cover multiple modalities. Many of the subtasks are considered standard tasks in their domain. Each subtask includes 1000 samples: for each sample the model has to select the correct answer among 1000 candidates, except some classification tasks (e.g. CIFAR-10) contain fewer candidates. We report the mean top-1 accuracy over all subtasks within each category.

**Visual Grounding with OpenImages.** We leverage OpenImages (Kuznetsova et al., 2020) to create a new task where the goal is to select the correct crop of an image given a query text. For **OI-Crop** the query text is the name of an object in the image. As candidates, we include five crops of other objects from the query image and five crops of the same object from other images (in contrast to VG-Crop which uses only crops of the same image). For **OI-Pos**, the model has to select the correct crop of an object from an image that contains this object exactly twice, given an instruction to select the left/right instance. The retrieval pool includes crops from the outer parts of the query image as decoy. We give a detailed description of the data collection method in App. A.4, which results in 1046 and 2546 samples respectively (see Fig. 2).

**VG-Crop.** We use 1574 validation samples from VG-Crop (Sec. 4.2), using all existing regions in each sample as the respective candidates (see Fig. 2). We prune regions that significantly overlap (IoU over 0.3) to facilitate unambiguous tasks. The size of the retrieval pool depends on the amount of available region descriptions, with an average of 15.9 images.

**CC3M-TGIT.** We use the validation split of CC3M-TGIT to test models on retrieving the correctly transformed image. For *crop* and *rotation* the retrieval

Table 2: **Number of samples per dataset.** We show the number of samples (in million) of the training datasets described in Sec. 4, and the combination of modalities that each dataset exhibits.

| Setting | CC | TGIT | CC3M-VQA | VG-VQA | VG-Crop | HQ-Edit |
|---|---|---|---|---|---|---|
| | I − T | IT − I | IT − T | IT − T | IT − I | IT − I(T) |
| CC3M | 2.6 | 0.3 | 2.4 | 0.7 | 5.4 | 0.3 |
| CC12M | 10.6 | 0.3 | 2.4 | 1.4 | 5.4 | 0.3 |

Table 3: **Evaluation on embedding tasks.** We report accuracy of late fusion baselines (SigLIP$_{SF}$, SigLIP$_{MLF}$) and FuseLIP models, trained on CC3M or CC12M plus multimodal data. Our FuseLIP-B achieves the best results across nearly all tasks, despite having fewer trainable parameters than SigLIP-B. We observe large margins on OI-Pos and TGIT, even for FuseLIP-S, indicating that early fusion better captures image-text relations. $^\star$*zero-shot*

| Training | Model | Classification$^\star$ | VQA$^\star$ | Retrieval$^\star$ | Grounding$^\star$ | ImageNet$^\star$ | VG-Crop | OI-Crop$^\star$ | OI-Pos$^\star$ | TGIT |
|---|---|---|---|---|---|---|---|---|---|---|
| CC3M +MM | SigLIP-S$_{SF}$ | 21.5 | 12.7 | 13.0 | 74.8 | 8.8 | 52.0 | 55.2 | 45.4 | 57.3 |
| | SigLIP-S$_{MLF}$ | 18.0 | 14.2 | 12.7 | 74.2 | 10.2 | 53.0 | 66.2 | 46.9 | 67.2 |
| | SigLIP-B$_{SF}$ | 22.2 | 13.6 | 13.4 | 77.2 | 10.3 | 55.1 | 56.9 | 45.9 | 56.6 |
| | SigLIP-B$_{MLF}$ | 19.5 | 14.8 | 13.9 | 76.9 | 12.2 | 55.4 | **68.4** | 47.4 | 69.4 |
| | FuseLIP-S | 18.5 | 15.9 | 11.2 | 70.8 | 13.5 | 49.6 | 59.8 | 53.9 | 79.0 |
| | FuseLIP-B | **23.3** | **17.5** | **15.0** | **82.4** | **18.1** | **55.8** | 68.1 | **70.8** | **94.3** |
| CC12M +MM | SigLIP-S$_{SF}$ | 30.4 | 16.2 | 23.8 | 74.2 | 21.4 | 57.1 | 60.1 | 47.1 | 66.0 |
| | SigLIP-S$_{MLF}$ | 28.5 | 16.9 | 23.2 | 72.7 | 25.5 | 58.8 | 72.2 | 46.6 | 81.0 |
| | SigLIP-B$_{SF}$ | **31.5** | 17.0 | 23.8 | 72.7 | 25.4 | 58.0 | 63.2 | 47.3 | 67.1 |
| | SigLIP-B$_{MLF}$ | 30.3 | 16.8 | 23.2 | 73.4 | 28.8 | **61.5** | **74.0** | 48.9 | 78.1 |
| | FuseLIP-S | 25.2 | 18.2 | 20.1 | 75.2 | 26.0 | 53.5 | 64.7 | 61.5 | 90.6 |
| | FuseLIP-B | 31.2 | **19.8** | **26.2** | **82.3** | **32.7** | **61.5** | 71.3 | **68.9** | **94.2** |

pools consists of all possible transformations (9 and 18), for *jitter* we use 10, for *flip* the original and horizontally/vertically flipped images (3), and for *colorize* the original and the target sample (2). Each subtask is evaluated on 1000 samples.

**ImageNet.** We evaluate zero-shot classification on the full ImageNet-1k (Deng et al., 2009) validation set. Following CLIP (Radford et al., 2021), we use the ensemble of 80 prompt templates per class, average the corresponding text embeddings, and assign each image to the class with highest cosine similarity. This differs from the ImageNet evaluation included in MMEB, which uses only 1k validation samples and a single text prompt template per class ("a photo of a class").

## 5.2 Main results

Table 3 reports the performance of the various models on the evaluation tasks detailed above. First, FuseLIP-B achieves the best results across nearly all tasks and training data configurations, often with large margin. It attains highest scores on 8 respectively 7 out of 9 benchmarks in the CC3M+MM and CC12M+MM training configurations. While the total parameters of FuseLIP-B are similar to B-sized baselines, it has significantly fewer trainable ones (Table 1). Notably, the non-trainable parameters come from the frozen image tokenizer, which is trained for image reconstruction and does not contribute directly to image-text alignment. We emphasize that this comparison concerns trainable parameters and hence optimization and training cost. At inference time, the frozen tokenizer is still part of FuseLIP and must be evaluated for image-containing inputs (see Table 7 for a runtime comparison).

Second, FuseLIP-S shows a more mixed picture. While FuseLIP-S has a similar total parameter count to the S-sized baselines, it has substantially fewer trainable parameters because the image tokenizer is frozen (see Table 1). This reduced trainable capacity, combined with the lower fidelity image tokenizer, likely makes FuseLIP-S less competitive on some standard embedding tasks, such as Classification and Retrieval. Nevertheless, FuseLIP-S already improves over the S- and B-sized late-fusion baselines on several tasks that require joint image-text reasoning, in particular VQA, OI-Pos, and TGIT. Scaling to FuseLIP-B, which uses both a stronger tokenizer and a larger transformer encoder, yields the strongest overall results. As FuseLIP significantly outperforms late fusion models, especially score fusion, on CC3M-TGIT, we analyze this key result in more detail below. Overall, these results suggest that early fusion of discrete tokens, even with a single encoder, is highly effective for aligning multimodal representations.

| Query | SigLIP-S$_{SF}$ | SigLIP-S$_{MLF}$ | SigLIP-B$_{SF}$ | SigLIP-B$_{MLF}$ | FuseLIP-S | FuseLIP-B |
|-------|------------------|-------------------|------------------|-------------------|-----------|-----------|
| Horizontal flip | ✗ | ✗ | ✗ | ✓ | ✓ | ✓ |

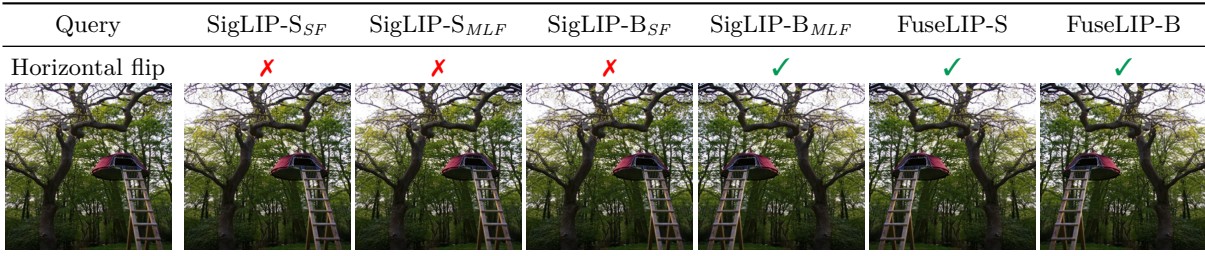

Figure 3: **TGIT evaluation examples.** We illustrate the "horizontal flip" task on models trained on CC12M+MM. While several baselines select the original image, only SigLIP-B$_{MLF}$ and both FuseLIP variants return the correctly flipped image. More examples are shown in Fig. 5.

**Why early fusion helps to understand text-guided transformations.** Our models achieve the largest improvements over baselines on CC3M-TGIT, where even FuseLIP-S outperforms SigLIP-B$_{MLF}$ by 9–10% and SigLIP-B$_{SF}$ by 22–24%. To analyze this effect, Table 4 presents a breakdown of performance across individual tasks in CC3M-TGIT. The advantage of FuseLIP emerges specifically in tasks requiring identification of the correct image after cropping, rotation, or flipping (horizontal or vertical). Unlike the baselines, FuseLIP solves these tasks almost perfectly (see qualitative examples in Figs. 3 and 5). We note that TGIT is evaluated on a held-out evaluation set, but the same transformation families are also part of the training mixture. Thus, these results measure the learnability of text-guided transformations rather than generalization to unseen transformation families. The improvement over baselines likely stems from the nature of these tasks, which rely on capturing the visual structure rather than semantic content. The unimodal encoders tend to extract semantic information which is used to align different inputs in the latent space. It is reasonable to hypothesize that features at deeper layers contain higher amount of semantic information, at the expense of other aspects such as visual information (Dorszewski et al., 2025). Moreover, attending to both image and text is crucial to solve these tasks, as no shortcut can be learned by just looking at input images and target images or input text and target images. Late fusion models likely have limited access to the information necessary to solve the task, while our early fusion approach can easily learn it. This is further supported by the comparison between late fusion baselines: SigLIP$_{MLF}$, which uses a learnable fusion network, substantially outperforms SigLIP$_{SF}$ on 3 out of 5 transformations, while achieving comparable performance on the remaining ones. Thus, learnable late fusion can improve over simple score fusion in several cases, but remains limited compared to early fusion. This interpretation can also explain the better results of FuseLIP compared to the baselines on OI-Pos, where the model needs to distinguish left and right instances of the same object.

Table 4: **Breakdown over TGIT tasks.** We report accuracy (%) on five text-guided transformations for models trained on CC12M+MM. Only FuseLIP variants solve *Crop*, *Rotate* and *Flip*.

| Model | Crop | Rotate | Flip | Jitter | Color |
|-------|------|--------|------|--------|-------|
| SigLIP-S$_{SF}$ | 49.7 | 84.7 | 15.6 | 80.7 | 99.4 |
| SigLIP-S$_{MLF}$ | 75.8 | 83.3 | 56.4 | 89.7 | 100.0 |
| SigLIP-B$_{SF}$ | 48.5 | 84.6 | 18.5 | 84.2 | 99.7 |
| SigLIP-B$_{MLF}$ | 55.0 | 85.8 | 58.8 | 90.8 | 99.9 |
| FuseLIP-S | 97.4 | 81.8 | 92.1 | 82.2 | 99.6 |
| FuseLIP-B | 99.7 | 94.7 | 89.2 | 87.7 | 99.6 |

**Fine-tuning on MMEB.** We finetune all models that were trained on CC12M+MM on the training split of the MMEB dataset (Jiang et al., 2025) for 10 epochs and report the results in Table 5. All models improve on the MMEB tasks *Classification, VQA, and Retrieval*, while suffering from performance drops in *Grounding*, which is likely due to the training set holding only few grounding samples. Models improve on ImageNet since it is part of the MMEB training split. Moreover, we observe unlearning on the VG-Crop, OI-Crop, and CC3M-TGIT tasks.

**Comparison to other models.** We compare to embedding models from other works in Table 5, namely a pre-trained SigLIP model (Zhai et al., 2023) and VLM2Vec (Jiang et al., 2025). We evaluate pre-trained SigLIP on multimodal embedding tasks via score fusion. This model has undergone much longer pre-training than our models, thus attains high performance on tasks that mainly require unimodal embeddings, such as

Table 5: **Fine-tuning on MMEB and comparison to other models.** We report evaluation results for models fine-tuned on the training split of MMEB (Jiang et al., 2025), using as initialization the models that were trained on CC12M+MM. Moreover, we compare to models from other works: pretrained SigLIP (Zhai et al., 2023) (much longer pretraining than our models) and VLM2Vec (Jiang et al., 2025) (much larger model and much longer pretraining than our models).

| Train Data | Model | Classif. | VQA | Retrieval | Grounding | ImNet | VG-Crop | OI-Crop | OI-Pos | TGIT |
|---|---|---|---|---|---|---|---|---|---|---|
| CC12M +MM →MMEB | SigLIP-S$_{SF}$ | 38.4 | 20.5 | 32.2 | 55.3 | 31.1 | 36.2 | 39.5 | 44.3 | 41.6 |
| | SigLIP-S$_{MLF}$ | 38.0 | 21.6 | 31.9 | 61.3 | 37.3 | 33.9 | 50.5 | 44.3 | 43.6 |
| | SigLIP-B$_{SF}$ | 40.4 | 22.0 | 34.6 | 55.1 | 36.6 | 36.7 | 41.7 | 45.4 | 43.7 |
| | SigLIP-B$_{MLF}$ | 39.8 | 22.1 | 34.1 | 61.9 | 41.8 | 36.4 | 47.5 | 43.0 | 39.9 |
| | FuseLIP-S | 33.6 | 22.6 | 27.2 | 62.8 | 33.2 | 32.4 | 47.0 | 52.9 | 50.9 |
| | FuseLIP-B | 40.4 | 24.9 | 34.9 | 73.3 | 41.9 | 36.4 | 51.8 | 53.1 | 54.8 |
| Pretrained | SigLIP-B/16 (Zhai et al., 2023) | 51.3 | 11.0 | 47.1 | 58.2 | 76.1 | 8.1 | 27.6 | 34.2 | 6.6 |
| Pretrained →MMEB | VLM2Vec Phi-3.5-V (Jiang et al., 2025) | 53.1 | 55.1 | 63.4 | 77.3 | 60.0 | 15.5 | 37.7 | 50.0 | 12.4 |
| − | Random Chance | 5.6 | 0.1 | 0.1 | 0.1 | 0.1 | 6.3 | 1.0 | 26.6 | 22.0 |

Table 6: **Ablations.** We report the effect of removing hard negatives (Sec. 4.3) or the MMM loss (Sec. 3.2) when training on CC3M+MM. Hard negatives are crucial for performance on VG-Crop, OI-Crop, OI-Pos, TGIT. The MMM loss can only be applied to FuseLIP and improves performance across all tasks.

| Model | Hard Neg. | $\mathcal{L}_{MMM}$ | Classif.$^\star$ | VQA$^\star$ | Retrieval$^\star$ | Grounding$^\star$ | ImNet$^\star$ | VG-Crop | OI-Crop$^\star$ | OI-Pos$^\star$ | TGIT |
|---|---|---|---|---|---|---|---|---|---|---|---|
| SigLIP-S$_{SF}$ | ✗ | - | 19.5 | 13.2 | 11.3 | 80.6 | 8.6 | 23.4 | 39.1 | 40.5 | 11.5 |
| | ✓ | - | 21.5 | 12.7 | 13.0 | 74.8 | 8.8 | 52.0 | 55.2 | 45.4 | 57.3 |
| SigLIP-S$_{MLF}$ | ✗ | - | 17.1 | 13.5 | 11.4 | 76.5 | 9.7 | 34.1 | 46.6 | 43.0 | 22.1 |
| | ✓ | - | 18.0 | 14.2 | 12.7 | 74.2 | 10.2 | 53.0 | 66.2 | 46.9 | 67.2 |
| FuseLIP-S | ✗ | ✓ | 20.0 | 16.5 | 11.9 | 76.2 | 15.1 | 35.0 | 51.0 | 51.7 | 18.5 |
| | ✓ | ✗ | 17.8 | 15.7 | 10.5 | 66.2 | 12.3 | 46.6 | 58.8 | 50.7 | 83.8 |
| | ✓ | ✓ | 18.5 | 15.9 | 11.2 | 70.8 | 13.5 | 49.6 | 59.8 | 53.9 | 79.0 |
| FuseLIP-B | ✗ | ✓ | 23.3 | 17.5 | 14.6 | 82.5 | 18.4 | 38.8 | 52.5 | 45.3 | 13.6 |
| | ✓ | ✗ | 21.0 | 16.8 | 14.3 | 81.3 | 16.3 | 53.7 | 63.3 | 68.0 | 88.4 |
| | ✓ | ✓ | 23.3 | 17.5 | 15.0 | 82.4 | 18.1 | 55.8 | 68.1 | 70.8 | 94.3 |

Classification and ImageNet. However, our models are better on multimodal tasks. Note that the MMEB results of the pre-trained SigLIP are higher than reported in the online leaderboard,[1] since we use more suitable prompts as described in App. A.4. Moreover, we evaluate a VLM2Vec model (Jiang et al., 2025) that is based on the Phi-3.5-V large vision-language model and fine-tuned on MMEB. Notably, this model has much more parameters (4.15B, i.e. over 25x that of FuseLIP-B) and has undergone much longer pre-training. Expectedly, this model outperforms our MMEB-fine-tuned models on MMEB evaluation tasks. However, both FuseLIP-S and FuseLIP-B achieve higher performance on our VG-Crop, OI-Crop, OI-Pos, and CC3M-TGIT tasks, thus highlighting the effectiveness of early fusion on our challenging multimodal tasks. Both pre-trained SigLIP and VLM2Vec perform below random chance level on CC3M-TGIT, due to their bias towards selecting the original image rather than the transformed one from the retrieval pool.

### 5.3 Ablation on the importance of hard negatives and masked modeling loss

To better understand the contribution of some design choices in FuseLIP, we train models selectively removing the hard negatives from training (Sec. 4.3) and the masked modeling loss. Results are reported in Table 6. First, we see that not including hard negatives in the batch causes large drops in performance on VG-Crop,

---

[1] https://huggingface.co/spaces/TIGER-Lab/MMEB-Leaderboard

OI-Crop, OI-Pos, and especially CC3M-TGIT (e.g., for FuseLIP-B accuracy decreases from 94.3% to 13.6%). Moreover, adding the hard negatives does not affect performance on the other tasks for FuseLIP-B, and yields only a minor decrease for the smaller FuseLIP-S. Similar observations about the benefit of hard negatives hold true also for the late fusion baselines. Second, Table 6 illustrates the role of the MMM loss: it yields improvements across most tasks, in particular for the larger FuseLIP-B. However, even without the MMM loss FuseLIP-B remains substantially stronger than the baselines on most tasks, thereby confirming that it is not the sole driver of FuseLIP's gains. Rather, the MMM loss provides an additional improvement that is naturally supported by the shared discrete-token architecture. We provide additional evidence of the advantages given by the MMM loss in Table 19 in the Appendix.

Further ablations and properties of FuseLIP such as variance over random seeds for training, using discrete image tokens in late fusion, compositionality, and the modality gap are studied in Appendix B.

## 6 Discussion

The results of FuseLIP have several implications. First, it is possible to train a CLIP-like model (on unimodal or multimodal data) using a single encoder, in contrast to standard CLIP models that rely on separate text and image encoders. Second, our architecture, which inherently supports multimodal embeddings, enables seamless integration of contrastive and masked modeling objectives. This significantly simplifies the FLAVA training setup (see Sec. 3.2), showing that both objectives can be combined without requiring separate forward passes. Moreover, our models achieve stable training by using standard recipes. Third, our results highlight tasks, such as the text-guided transformations, where early fusion of modalities significantly outperforms late fusion. Since solving such tasks is part of a comprehensive multimodal encoder, we argue that early fusion is particularly promising for multimodal embeddings. Finally, we anticipate that FuseLIP can be naturally extended to new applications, e.g., encoding multiple images.

**Limitations.** Given limited computational resources, we could not test the effect of further scaling data and model size. Moreover, while our models are faster and require significantly less memory at training time, they are more expensive at inference than the baselines, but we expect the gap to be reduced at scale and thanks to the ongoing progress of image tokenizers.

## 7 Conclusion

We have introduced a novel approach to multimodal embedding models, designing an early fusion architecture with discrete image tokenization and a single encoder. Our simple training recipe combines contrastive and masked modeling objectives, while leveraging hard negative samples. The individual components as well as the final models are empirically validated on a variety of tasks. We demonstrate the benefits of early fusion in a controlled head-to-head comparison against late fusion baselines. Our methods for generating training datasets that are tailored for multimodal learning can be applied at scale. Moreover, our novel evaluation tasks are complementary to the existing benchmarks typically used for testing embedding models. Overall, we believe our approach and datasets can be valuable building blocks for future research on multimodal embedding models.

## Broader Impact

This work is methodological and aims to improve multimodal representation learning. It inherits potential societal risks from its training data setup. Models trained on CC3M, CC12M, and OpenImages may reproduce biases, stereotypes, and coverage imbalances present in these web-scale datasets. Moreover, our CC3M-VQA data is generated from captions using and LLM, and can therefore encode both caption biases and language-model priors. Finally, capabilities such as solving OI-Pos, which require distinguishing spatially localized objects may be relevant to applications with privacy concerns such as identification, localization, or tracking, when combined with other systems.

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

## Appendix

This appendix provides additional details and results to support the main text. In App. A we report details on the data creation, model implementation, training scheme, and evaluation tasks. In App. B we proceed to report complementary results, specifically regarding compositionality, the modality gap, training late fusion baselines with frozen encoders, training on unimodal data, and further ablations.

## A Implementation Details

### A.1 Data

**CC3M-TGIT.** In the following we give a detailed description of the transformations applied in CC3M-TGIT and CC12M-TGIT. *random cropping*: crop to a location of the image, specified as "upper left, upper center,. . . ", thus there are 9 possible crop locations. *random rotations*: randomly rotate the image clockwise or counter-clockwise at a random angle sampled uniformly from $(10°, 20°, . . . , 90°)$. *flipping*: flip the image vertically or horizontally. *colorization*: convert the image to grayscale and use the original colored image as the target. *grayscale*: convert the image to grayscale and use it as the target, use the original image as query. *jitter*: apply color-jittering by randomly adjusting brightness, contrast, and saturation by random factors, which are sampled uniformly between 0.3 and 2.0 and rounded to one decimal place.

**CC3M-VQA.** We generate VQA data from CC3M by querying an LLM to rewrite the given captions into question-answer pairs. To this end, we use Llama-3.1-8B-Instruct (Dubey et al., 2024) with a custom system prompt that guides the model to produce QA-pairs via rules and examples. The full system prompt is reported in Fig. 4. Notably, the model receives only the captions (not the images). In order to encourage variability in the generated VQA data, we use stochastic decoding by sampling the next token based on the predicted probabilities.

### A.2 Models

Following SigLIP (Zhai et al., 2023), we use bidirectional attention for all models. Moreover, we generally use a context length of 180 and mask out padding tokens. The baseline models are based on the ViT-S/16 and ViT-B/16 CLIP models as implemented in OpenCLIP (Cherti et al., 2023). We present links and licenses for all assets used in this paper in Table 9.

**SigLIP$_{SF}$.** For the score-level fusion of SigLIP, we simply add the normalized features arithmetically, and normalize again after the addition.

**SigLIP$_{MLF}$.** Following the original implementation of MagicLens (Zhang et al., 2024), the fusion module of SigLIP-S$_{MLF}$ is a transformer, for which we scale the width and number of heads down to match those of ViT-S/16. Thereby, the fusion module has 4 layers at width 384 with 6 heads. For SigLIP-B$_{MLF}$ the fusion module has 4 layers at width 512 with 8 heads. The fusion module operates on the concatenated non-normalized image and text embeddings and is followed by an attention pooling layer. Whenever the model is queried without an image, we use a zero-tensor for the image embedding in the late fusion stage.

Table 7: **Memory and runtime comparison.**

| | Training | | Inference | |
|---|---|---|---|---|
| | Memory [GB] | Time [ms] | Memory [GB] | Time [ms] |
| SigLIP-S$_{SF}$ | 18.7 | 315 | 0.7 | 100 |
| SigLIP-S$_{MLF}$ | 18.8 | 328 | 0.7 | 104 |
| SigLIP-B$_{SF}$ | 32.2 | 595 | 1.5 | 199 |
| SigLIP-B$_{MLF}$ | 32.3 | 612 | 1.5 | 202 |
| FuseLIP-S | 11.0 | 243 | 1.2 | 129 |
| FuseLIP-B | 14.1 | 425 | 2.3 | 271 |

**FuseLIP.** For FuseLIP-S we use the TiTok family of image tokenizers (Yu et al., 2024). In particular, for FuseLIP-S we use TiTok-S-128 and the subsequent transformer is based on the text transformer of S-sized CLIP (width of 384, 6 heads, 12 layers). For FuseLIP-B, we use the TiTok-BL-128-VQ image tokenizer (Yu

```
You are a helpful assistant, that generates a question and answer about an image that has
    a given caption. Rules:
1. For generating question/answer pairs, only use information that is evident from the
    caption.
2. Do not mention the word 'caption' in the question or answer.
3. Answers should be at least a couple words long (not single word).
4. Don't start every question with "What"
5. Respond in the format: Question: <question>
Answer: <answer>
Examples:
Caption: A group of friends are having a barbecue in the backyard.
Question: Where is the barbecue taking place?
Answer: In the backyard.
Caption: A child is playing with a toy airplane on the floor.
Question: Which toy is the child holding?
Answer: A toy airplane.
Caption: A girl with a smartphone is lying on the sofa.
Question: Where is the girl in the image located?
Answer: On the sofa.
Caption: A golden retriever is running through a field of flowers.
Question: What is the dog doing?
Answer: Running through a field of flowers.
```

Figure 4: **System prompt for generating question-answer pairs.** We use Llama-3.1-8B-Instruct to generate VQA samples from image-text pairs of CC3M as described in Sec. 4.2.

et al., 2024) followed by the text transformer of B-sized CLIP (width of 512, 8 heads, 12 layers). We do not use the generators that were trained along with the image tokenizers.

**Memory requirements and runtime.** In Table 7 we show a comparison of the memory allocation and runtime at training and at inference time of the models considered in this paper. To this end we do forward passes at batch size 128 of two multimodal samples (image+text) and average this over 10 repetitions. The FuseLIP models profit at training time from fewer trainable parameters and exhibit significantly lower memory allocation than the baselines. This means that our models can be trained with the same batch size on fewer GPUs and thus save resources. As a drawback, the forward pass of FuseLIP is more expensive at inference since images are processed sequentially by both tokenizer and encoder. However, we expect this overhead to be reduced when scaling up the size of the encoder, which will dominate inference cost. For retrieval applications, embeddings of target images can typically be precomputed during indexing, so the tokenizer and encoder cost for the retrieval database is amortized.

### A.3  Training

**Batch composition.** We make sure that hard negatives are present in each batch by the sampling strategy outlined in Sec. 4.3. Here we give a detailed account of the amount of hard negatives sampled for each transformation in CC3M-TGIT and CC12M-TGIT. *crop*: we take all 9 possible crops. *rotate*: we take 3 randomly selected rotations. *jitter*: we take 3 samples. *flip*: we take horizontal flip and vertical flip, both applied once from the original image and once from the flipped image, yielding 4 samples. *colorize-grayscale*: for each colorization sample we also take the to-grayscale sample, and vice-versa, yielding 2 samples.

Table 8: **Hyperparameters for training.**

| Parameter | CC3M+MM | CC12M+MM |
|---|---|---|
| Epochs | 8 | 16 |
| Optimizer | AdamW | AdamW |
| Batch size | 2048 | 2048 |
| Weight decay | 1.0 | 0.5 |
| AdamW $\beta_1, \beta_2$ | 0.9, 0.98 | 0.9, 0.98 |
| AdamW $\epsilon$ | $1 \times 10^{-8}$ | $1 \times 10^{-8}$ |
| Learning rate | $1 \times 10^{-3}$ | $1 \times 10^{-3}$ |
| Learning rate schedule | cosine | cosine |
| Warmup steps | 12000 | 12000 |
| $\ell_2$-gradient clipping | 1.0 | 1.0 |
| Context length | 180 | 180 |
| Image resolution | 256 | 256 |

Table 9: **Assets used in this paper.**

| Asset | Link | License |
|---|---|---|
| OpenCLIP | https://github.com/mlfoundations/open_clip | MIT |
| TiTok | https://github.com/bytedance/1d-tokenizer | Apache-2.0 |
| Llama-3.1-8B-Instruct | https://huggingface.co/meta-llama/Llama-3.1-8B-Instruct | see Link |
| VLM2Vec | https://huggingface.co/TIGER-Lab/VLM2Vec-Full | Apache 2.0 |
| CC3M | https://huggingface.co/datasets/pixparse/cc3m-wds | see Link |
| CC12M | https://huggingface.co/datasets/pixparse/cc12m-wds | see Link |
| MMEB | https://huggingface.co/datasets/TIGER-Lab/MMEB-eval | Apache 2.0 |
| VisualGenome | https://homes.cs.washington.edu/~ranjay/visualgenome/index.html | CC BY 4.0 |
| OpenImages | https://storage.googleapis.com/openimages/web/index.html | CC BY 4.0 |
| SugarCrepe | https://github.com/RAIVNLab/sugar-crepe | MIT |

**Hyperparameters.** The hyperparameters used for training on CC3M+MM and CC12M+MM are reported in Table 8. We use $\ell_2$-norm gradient clipping for all models as we observed that this stabilizes multimodal training. We select AdamW (Loshchilov and Hutter, 2018) as the optimizer.

### A.4 Evaluation tasks

**Massive Multimodal Embedding Benchmark (MMEB).** MMEB (Jiang et al., 2025) is a multimodal embedding benchmark consisting of 36 datasets that are split into the categories Classification, VQA, Retrieval, and Grounding. Each dataset consists of 1000 samples, and for each sample there are 1000 candidates, except for classification datasets that contain less classes. Only one of the candidates is correct. Since MMEB was created to train and evaluate VLM2Vec (Jiang et al., 2025), an embedding model that is based on an autoregressive large multimodal model, it contains prompts in instruction format. As our models are not trained on instruction-following data, we remove this part of the prompts (except when evaluating the VLM2Vec model in Table 5).

**OpenImages-Crop (OI-Crop.)** To create the OI-Crop task, we start with the OpenImages dataset and first remove all bounding boxes that are very small (less than 50 pixels), very large (larger than 0.9 relative size in either dimension), that have high aspect ratio (larger than 1.5). Then we remove boxes with labels that appear less than ten times in total. Next, we filter out samples that have less than five uniquely labelled bounding boxes, then we drop for each sample bounding boxes that are strongly overlapping (IoU larger than 0.6). Finally, we gather for each sample a label with the corresponding bounding box, and collect as negatives four bounding boxes from the same image (with different label) and five bounding boxes from other images with the same label. Since the distribution of label names is heavily skewed, we make sure to get each label name at maximum five times as a query. In total this procedure yields 1046 samples, some are shown in Fig. 2.

**OpenImages-Position (OI-Pos).** For OI-Pos we select OpenImages images that contain an object exactly twice. We consider 34 object classes, chosen so that identification of individual instances is generally possible (e.g. for "Window" this is not the case, as many images with windows contain additional windows that are not labelled). We drop images where the bounding boxes overlap significantly in horizontal direction (if the horizontal center of either box overlaps the other box). Next, we obtain one negative bounding box each from the left and right border of the image. If this would cause overlap to the positive box, we obtain boxes from the top or bottom. If there is also not enough space (we enforce min. size of 30x30 pixels), the retrieval pool could contain less than four samples. Then we restrict each label to 100 samples. The query is then the image with the text "The {object_name} on the left/right", and the retrieval pool consists of the two object crops and the two border crops. In total this task contains 2546 samples. See Fig. 2 for some examples.

**ImageNet.** We evaluate on the full ImageNet-1k validation set (Deng et al., 2009), and use the ensemble of OpenAI prompt templates (Radford et al., 2021). In contrast, the ImageNet evaluation as part of MMEB uses only a single prompt and 1k samples.

Table 10: **Variance of training runs.** We report mean and standard deviation over 3 training runs on CC3M+MM with different random seeds. FuseLIP maintains a clear advantage over SigLIP-B$_{MLF}$ on most benchmarks. *⋆zero-shot*

| | Classif.⋆ | VQA⋆ | Retrieval⋆ | Grounding⋆ | ImageNet ⋆ | VG-Crop | OI-Crop⋆ | OI-Pos⋆ | TGIT |
|---|---|---|---|---|---|---|---|---|---|
| SigLIP-B$_{MLF}$ | $20.2_{\pm0.8}$ | $14.3_{\pm0.4}$ | $14.0_{\pm0.2}$ | $75.7_{\pm1.0}$ | $12.3_{\pm0.4}$ | $\mathbf{55.9}_{\pm0.8}$ | $\mathbf{67.8}_{\pm0.8}$ | $47.5_{\pm0.2}$ | $70.9_{\pm1.7}$ |
| FuseLIP-B | $\mathbf{23.0}_{\pm0.5}$ | $\mathbf{17.2}_{\pm0.5}$ | $\mathbf{15.0}_{\pm0.8}$ | $\mathbf{82.1}_{\pm0.6}$ | $\mathbf{17.2}_{\pm1.5}$ | $55.6_{\pm1.9}$ | $67.4_{\pm1.2}$ | $\mathbf{69.0}_{\pm1.5}$ | $\mathbf{91.2}_{\pm5.5}$ |

Table 11: **Evaluation on SugarCrepe.** FuseLIP outperforms the baselines on most compositionality tasks.

| Train Data | Model | Replace | | | | Swap | | | Add | | |
|---|---|---|---|---|---|---|---|---|---|---|---|
| | | Object | Attribute | Relation | **Mean** | Object | Attribute | **Mean** | Object | Attribute | **Mean** |
| CC3M +MM | SigLIP-S$_{SF}$ | 67.4 | 69.0 | 56.8 | 64.4 | 51.8 | 54.5 | 53.2 | 46.5 | 46.1 | 46.3 |
| | SigLIP-S$_{MLF}$ | 71.8 | 69.8 | **59.9** | 67.2 | 52.2 | 59.5 | 55.8 | 66.8 | 60.1 | 63.4 |
| | SigLIP-B$_{SF}$ | 70.4 | 71.2 | 59.5 | 67.0 | 47.4 | 58.4 | 52.9 | 48.2 | 46.1 | 47.1 |
| | SigLIP-B$_{MLF}$ | 75.8 | 70.7 | 59.5 | 68.7 | **58.0** | 59.3 | 58.6 | 69.7 | 62.3 | 66.0 |
| | FuseLIP-S | 74.2 | 69.5 | 58.8 | 67.5 | 50.6 | 62.9 | 56.8 | 71.3 | 60.8 | 66.1 |
| | FuseLIP-B | **78.8** | **71.3** | 59.4 | **69.8** | 56.3 | **64.0** | **60.1** | **73.8** | **66.8** | **70.3** |
| CC12M +MM | SigLIP-S$_{SF}$ | 79.6 | 74.4 | 65.1 | 73.0 | 62.0 | 63.7 | 62.8 | 55.9 | 55.4 | 55.6 |
| | SigLIP-S$_{MLF}$ | 85.4 | 78.3 | **69.7** | 77.8 | 62.4 | 65.8 | 64.1 | 78.8 | 72.4 | 75.6 |
| | SigLIP-B$_{SF}$ | 82.4 | 76.0 | 64.2 | 74.2 | 63.3 | 63.5 | 63.4 | 54.4 | 52.0 | 53.2 |
| | SigLIP-B$_{MLF}$ | **88.3** | 78.3 | 67.8 | 78.1 | **66.5** | 69.4 | 68.0 | 79.0 | **73.4** | 76.2 |
| | FuseLIP-S | 83.4 | 77.4 | 66.4 | 75.8 | 60.0 | 68.0 | 64.0 | 76.6 | 71.0 | 73.8 |
| | FuseLIP-B | 87.6 | **78.9** | 69.6 | **78.7** | **66.5** | **69.8** | **68.2** | **82.4** | 72.0 | **77.2** |

# B  Additional Results

**Variance of training runs.** To assess the stability of our results, we repeat training of SigLIP-B$_{MLF}$ and FuseLIP-B on CC3M+MM with 3 different random seeds, reporting mean and standard deviation in Table 10. FuseLIP-B outperforms SigLIP-B$_{MLF}$ on most benchmarks, with gains that exceed the observed run-to-run variation. The only exceptions are VG-Crop and OI-Crop, where both methods perform comparably.

**Compositionality.** The SugarCrepe benchmark (Hsieh et al., 2023) measures how well models capture compositional concepts in text and images. This benchmark considers modifying objects, attributes, and relations of sentences via replacing, swapping, and adding. To test how the different approaches to multimodal embeddings influence the compositionality ability, we report their performance on SugarCrepe in Table 11. In both training setups (CC3M+MM and CC12M+MM), our FuseLIP-B attains the highest mean performances across compositionality categories. This result suggests another potential benefit of early fusion.

**Modality gap.** The modality gap in vision-language models with unimodal embedding, i.e. images and text are mapped to different regions of the latent space, is a well-known phenomenon (Liang et al., 2022; Shi et al., 2023). With multimodal embedding models we can also study the relative position of the representations of multimodal inputs. In Table 12 we compute the modality gap as defined by Liang et al. (2022):

$$\left\| \frac{1}{n}\sum_{i=1}^{n} f(z_i^1) - \frac{1}{n}\sum_{i=1}^{n} f(z_i^2) \right\|_2,$$

where $z_i^1$ and $z_i^2$ are inputs from different modalities. We select datasets representing the four combinations of data modalities spanned by the evaluation tasks. These datasets are sub-tasks of MMEB (Jiang et al., 2025). We observe that the results are similar across fusion models.

Table 13: **Ablation of the image tokenizer and transformer sizes.** We vary the image tokenizer and transformer sizes for models trained on CC3M+MM. While scaling the image tokenizer drives performance, scaling up the transformer also yields further improvements. The highlighted rows correspond to FuseLIP-S and FuseLIP-B respectively. $^\star$*zero-shot*

| Tokenizer | Transformer | Classif.$^\star$ | VQA$^\star$ | Retrieval$^\star$ | Grounding$^\star$ | ImageNet $^\star$ | VG-Crop | OI-Crop$^\star$ | OI-Pos$^\star$ | TGIT |
|---|---|---|---|---|---|---|---|---|---|---|
| S | S | 18.5 | 15.9 | 11.2 | 70.8 | 13.5 | 49.6 | 59.9 | 53.9 | 78.5 |
| S | B | 19.4 | 16.4 | 12.0 | 72.7 | 14.6 | 48.3 | 58.4 | 55.5 | 91.5 |
| B | S | 22.1 | 16.6 | 14.3 | 81.6 | 16.1 | 55.4 | 66.2 | 68.0 | 95.6 |
| B | B | 23.3 | 17.5 | 15.1 | 82.5 | 18.1 | 55.8 | 68.1 | 70.8 | 93.9 |

Interestingly, the gap between multimodal (IT) and unimodal samples (T or I) are larger than between unimodal inputs from different modalities (I – T), while the gap for IT – IT data is very small as expected. Finally, we test the modality gap at initialization, i.e., with random encoders: early fusion (FuseLIP-S), with more shared weights, yields smaller gaps than late fusion, but this difference disappears after training.

**Ablation of the image tokenizer.** We employ two image tokenizers in our work: TiTok-S as part of FuseLIP-S, and TiTok-B as part of FuseLIP-B. Yu et al. (2024) report reconstruction FIDs on ImageNet of 1.71 and 1.49 for

Table 12: **Modality gap.** We report the modality gap (as defined by Liang et al. (2022)) for models at initialization, and trained on CC12M+MM.

|  |  | ImNet | OK-VQA | RefCOCO | RefCOCO-M |
|---|---|---|---|---|---|
|  |  | I – T | IT – T | IT – I | IT – IT |
| init. | SigLIP-S$_{SF}$ | 1.24 | 0.86 | 0.65 | 0.02 |
|  | SigLIP-S$_{MLF}$ | 1.04 | 0.97 | 0.71 | 0.02 |
|  | FuseLIP-S | 0.72 | 0.68 | 0.19 | 0.01 |
| trained | SigLIP-S$_{SF}$ | 0.44 | 0.59 | 0.69 | 0.02 |
|  | SigLIP-S$_{MLF}$ | 0.39 | 0.46 | 0.64 | 0.02 |
|  | FuseLIP-S | 0.42 | 0.52 | 0.58 | 0.02 |

TiTok-S and TiTok-B respectively, demonstrating higher quality of the larger tokenizer. In our experiments we indeed observe that this improves the embedding performance of FuseLIP. In order to further ablate the role of the tokenizer, we train FuseLIP on CC3M+MM in two new configurations: (i) with TiTok-S and base-sized transformer encoder, and (ii) with TiTok-B and small-sized transformer encoder. We report the results in Table 13. We observe that the higher quality TiTok-B tokenizer already yields substantial improvements when used together with the small-sized transformer, confirming that tokenizer quality drives improvement. However, scaling up the transformer also contributes to improved performance for both tokenizers. This ablation compares tokenizer fidelity at a fixed image-token budget, since both TiTok-S and TiTok-B represent each image with 128 tokens. Thus, the results indicate that higher tokenizer fidelity improves downstream performance without increasing the sequence length processed by the encoder. We do not vary the number of image tokens in this work, and therefore do not claim a scaling law with respect to token budget. However, increasing the token budget would directly increase the encoder sequence length and hence inference cost.

**Late fusion of discrete image tokens.** In order to disentangle the effect of the pretrained image tokenizer, we train a SigLIP-B$_{MLF}$ baseline, where the image encoder receives discrete image tokens from TiTok-B. We compare this against SigLIP-B$_{MLF}$ and FuseLIP-B in Table 14, trained on CC3M+MM. We observe that this approach generally reduces performance compared to the standard SigLIP-B$_{MLF}$ baseline, thereby confirming that the pretraining of the image tokenizer is not the main driver of FuseLIP performance, but rather its early fusion approach.

**Training SigLIP$_{MLF}$ with frozen image and text encoders.** For comparison, we test the performance of SigLIP$_{MLF}$ when trained with frozen pre-trained image and text encoder backbones. In order to compare models trained at similar scale, we pre-train unimodal SigLIP models on CC3M and use them as frozen initialization for SigLIP$_{MLF}$ training on CC3M+MM, which we denote as SigLIP-frozen$_{MLF}$. Results are reported in Table 15. Compared to SigLIP$_{MLF}$ trained multimodally from scratch, we observe that the frozen initialization helps on some tasks (Classification, Retrieval), while significantly degrading performance on

Table 14: **Baseline with image tokenizer.** In order to disentangle the effect of the pretrained image tokenizer, we train on CC3M+MM a SigLIP-B$_{MLF}$ baseline that receives image tokens from TiTok-B. We observe that this generally reduces performance compared to the standard SigLIP-B$_{MLF}$ baseline, thereby confirming that the pretraining of the image tokenizer is not the main driver of FuseLIP performance, but rather its early fusion approach. $^\star$*zero-shot*

|  | Classif.$^\star$ | VQA$^\star$ | Retrieval$^\star$ | Grounding$^\star$ | ImageNet $^\star$ | VG-Crop | OI-Crop$^\star$ | OI-Pos$^\star$ | TGIT |
|---|---|---|---|---|---|---|---|---|---|
| SigLIP-B$_{MLF}$ | 19.5 | 14.8 | 13.9 | 76.9 | 12.2 | 55.4 | **68.4** | 47.4 | 69.4 |
| TiTok-SigLIP-B$_{MLF}$ | 18.6 | 14.2 | 11.8 | 68.2 | 12.1 | 51.1 | 58.8 | 47.3 | 69.3 |
| FuseLIP-B | **23.3** | **17.5** | **15.0** | **82.4** | **18.1** | **55.8** | 68.1 | **70.8** | **94.3** |

Table 15: **SigLIP$_{MLF}$ with frozen backbones.** We train SigLIP$_{MLF}$ with frozen image and text encoder backbones (pre-trained on CC3M) on CC3M+MM. These models are outperformed by the models trained from scratch on most tasks, on some significantly. $^\star$*zero-shot*

| Model | Classification$^\star$ | VQA$^\star$ | Retrieval$^\star$ | Grounding$^\star$ | ImageNet $^\star$ | VG-Crop | OI-Crop$^\star$ | OI-Pos$^\star$ | TGIT |
|---|---|---|---|---|---|---|---|---|---|
| SigLIP-S$_{MLF}$ | 18.0 | 14.2 | 12.7 | 74.2 | 10.2 | 53.0 | 66.2 | 46.9 | 67.4 |
| SigLIP-S-frozen$_{MLF}$ | 21.2 | 12.8 | 12.7 | 72.6 | 8.4 | 51.5 | 56.7 | 45.6 | 60.7 |
| SigLIP-B$_{MLF}$ | 19.5 | 14.8 | 13.9 | 76.9 | 12.2 | 55.4 | 68.4 | 47.4 | 69.4 |
| SigLIP-B-frozen$_{MLF}$ | 21.1 | 13.3 | 14.9 | 57.5 | 17.9 | 41.1 | 46.9 | 45.2 | 53.3 |

several others (Grounding, VG-Crop, OI-Crop, OI-Pos, TGIT). We expect larger improvements when using a SigLIP model pre-trained on larger scale data as initialization. However, such comparison would not be fair, as the pre-trained model would have undergone a significant amount of image-text alignment. In contrast, the image tokenizer used in FuseLIP has only been trained for image compression on ImageNet (no textual alignment).

**Loss ablation.** As a sanity check, we report performance of FuseLIP-S trained on CC3M+MM with only the masked multimodal modeling (MMM) loss in Table 16. We observe that only using the MMM loss yields far worse performance, highlighting that the contrastive SigLIP loss is necessary to properly train the FuseLIP models.

**Performance on unimodal tasks.** While this paper focusses on multimodal embedding tasks, we also compare the performance of late and early fusion approaches on unimodal embedding tasks. These are tasks that do not require embedding image+text into a single feature vector, but rather into separate vectors, e.g. ImageNet classification. The Classification and Retrieval subtasks of MMEB contain several tasks that are unimodal, in addition to our ImageNet evaluation. In Table 17, we report the mean performance of the models on these unimodal tasks. While FuseLIP-S trails some late-fusion baselines at CC12M+MM scale, our FuseLIP-B (same number of trainable parameters as SigLIP-S) consistently outperforms all baselines across tasks, indicating that early fusion does not sacrifice unimodal embedding ability.

Table 16: **Loss ablation.** We compare training FuseLIP with only the masked multimodal modeling (MMM) loss. We observe that this yields weak performance, confirming that the employed contrastive loss is necessary.

| Model | Classification$^\star$ | VQA$^\star$ | Retrieval$^\star$ | Grounding$^\star$ | ImNet$^\star$ | VG-Crop | OI-Crop$^\star$ | OI-Pos$^\star$ | TGIT |
|---|---|---|---|---|---|---|---|---|---|
| FuseLIP-S-only-MMM | 8.5 | 0.2 | 2.6 | 30.2 | 0.1 | 6.3 | 20.2 | 31.2 | 5.6 |
| FuseLIP-S | 18.5 | 15.9 | 11.2 | 70.8 | 13.5 | 49.6 | 59.9 | 53.9 | 78.5 |

Table 17: **Unimodal tasks.** We report average performances over unimodal tasks, split over ImageNet, MMEB unimodal classification subtasks, and MMEB unimodal retrieval subtasks. $^{\star}$*zero-shot*

| Train Data | Model | ImageNet$^{\star}$ | Unimodal Classification$^{\star}$ | Unimodal Retrieval$^{\star}$ |
|---|---|---|---|---|
| CC3M +MM | SigLIP-S$_{SF}$ | 8.8 | 17.5 | 16.4 |
| | SigLIP-S$_{MLF}$ | 10.2 | 13.8 | 16.3 |
| | SigLIP-B$_{SF}$ | 10.3 | 18.2 | 17.0 |
| | SigLIP-B$_{MLF}$ | 12.2 | 15.8 | 18.1 |
| | FuseLIP-S | 13.5 | 14.3 | 14.3 |
| | FuseLIP-B | 18.1 | 19.9 | 18.9 |
| CC12M +MM | SigLIP-S$_{SF}$ | 21.5 | 27.8 | 28.8 |
| | SigLIP-S$_{MLF}$ | 25.5 | 27.4 | 28.9 |
| | SigLIP-B$_{SF}$ | 25.4 | 29.2 | 29.0 |
| | SigLIP-B$_{MLF}$ | 28.8 | 28.1 | 28.1 |
| | FuseLIP-S | 26.0 | 22.6 | 25.3 |
| | FuseLIP-B | 32.7 | 29.3 | 31.7 |

Table 18: **CC3M and CC12M unimodal training.** We report evaluation results for models trained on CC3M and CC12M without any multimodal embedding data.

| Train Data | Model | Classification$^{\star}$ | VQA$^{\star}$ | Retrieval$^{\star}$ | Grounding$^{\star}$ | ImageNet$^{\star}$ | VG-Crop$^{\star}$ | OI-Crop$^{\star}$ | OI-Pos $^{\star}$ | TGIT$^{\star}$ |
|---|---|---|---|---|---|---|---|---|---|---|
| CC3M | SigLIP-S | 20.7 | 4.0 | 15.7 | 39.0 | 19.5 | 9.8 | 26.6 | 34.8 | 7.7 |
| | FuseLIP-S | 16.7 | 1.1 | 8.1 | 22.0 | 15.4 | 6.5 | 19.2 | 31.5 | 7.6 |
| | FuseLIP-B | 21.4 | 2.6 | 13.1 | 39.0 | 21.7 | 6.3 | 20.9 | 32.3 | 5.8 |
| CC12M | SigLIP-S | 31.6 | 4.8 | 29.9 | 48.3 | 38.2 | 8.6 | 28.5 | 35.0 | 10.9 |
| | FuseLIP-S | 24.2 | 2.1 | 17.6 | 33.0 | 29.5 | 6.7 | 20.6 | 31.9 | 6.3 |
| | FuseLIP-B | 30.0 | 4.2 | 26.7 | 43.3 | 37.0 | 6.4 | 22.1 | 32.4 | 5.6 |

Table 19: **Contribution of the MMM loss.** Results on additional pre-training data.

| Train Data | Model | $\mathcal{L}_{\mathrm{MMM}}$ | Classification$^\star$ | VQA$^\star$ | Retrieval$^\star$ | Grounding$^\star$ | ImNet$^\star$ | VG-Crop | OI-Crop$^\star$ | TGIT |
|---|---|---|---|---|---|---|---|---|---|---|
| CC3M | FuseLIP-S | ✗ | 16.3 | 0.8 | 7.6 | 23.3 | 14.3 | 6.5 | 16.2 | 8.4 |
| | FuseLIP-S | ✓ | 16.7 | 1.1 | 8.1 | 22.0 | 15.4 | 6.5 | 19.2 | 7.6 |
| | FuseLIP-B | ✗ | 18.8 | 1.8 | 12.6 | 36.2 | 19.6 | 6.2 | 20.6 | 5.6 |
| | FuseLIP-B | ✓ | 21.4 | 2.6 | 13.1 | 39.0 | 21.7 | 6.3 | 20.9 | 5.8 |
| CC12M | FuseLIP-S | ✗ | 23.1 | 2.0 | 16.9 | 35.4 | 29.0 | 6.3 | 20.8 | 5.9 |
| | FuseLIP-S | ✓ | 24.2 | 2.1 | 17.6 | 33.0 | 29.5 | 6.7 | 20.6 | 6.3 |
| | FuseLIP-B | ✗ | 26.8 | 3.5 | 22.0 | 40.1 | 35.0 | 6.4 | 20.5 | 5.9 |
| | FuseLIP-B | ✓ | 30.0 | 4.2 | 26.7 | 43.3 | 37.0 | 6.4 | 22.1 | 5.6 |
| CC12M+MM | FuseLIP-S | ✗ | 23.5 | 17.7 | 19.2 | 69.9 | 24.4 | 53.3 | 63.1 | 89.3 |
| | FuseLIP-S | ✓ | 25.2 | 18.2 | 20.1 | 75.2 | 26.0 | 53.5 | 64.7 | 90.6 |
| | FuseLIP-B | ✗ | 28.3 | 19.1 | 23.8 | 81.8 | 30.8 | 59.5 | 70.6 | 95.8 |
| | FuseLIP-B | ✓ | 31.2 | 19.8 | 26.2 | 82.3 | 32.7 | 61.5 | 71.3 | 94.2 |

**Training on unimodal data.** As an ablation we train models on CC3M and CC12M only, without any multimodal embedding data, corresponding to the standard CLIP-like training setting. We train for 32 epochs on CC3M and for 30 epochs on CC12M. This yields a total number of seen samples of 93M and 327M respectively, which is similar to the training runs on multimodal data as described in Sec. 5. We evaluate on the same tasks as in the main paper. To this end, we use score fusion for SigLIP for any task that requires multimodal embeddings at evaluation time. We do not train a SigLIP$_{MLF}$ model in this setting, as the late fusion module would always receive a placeholder embedding for the missing modality at training time. We report the results in Table 18 and observe that performance on multimodal embeddings tasks is expectedly bad when compared to multimodal training. However, on unimodal embedding tasks the performance is similar (*Classification*) or improved (*ImageNet*). On these tasks, FuseLIP-B is slightly better than SigLIP-S when trained on CC3M, and slightly worse when trained on CC12M.

**Extended MMEB results.** We show a breakdown over all subtasks of MMEB for models trained on CC3M+MM in Table 20 and for models trained on CC12M+MM in Table 21.

**Qualitative examples.** We show qualitative examples of images retrieved by models trained on CC12M+MM on CC3M-TGIT in Fig. 5, OI-Crop and OI-Pos in Fig. 6, and VG-Crop in Fig. 7.

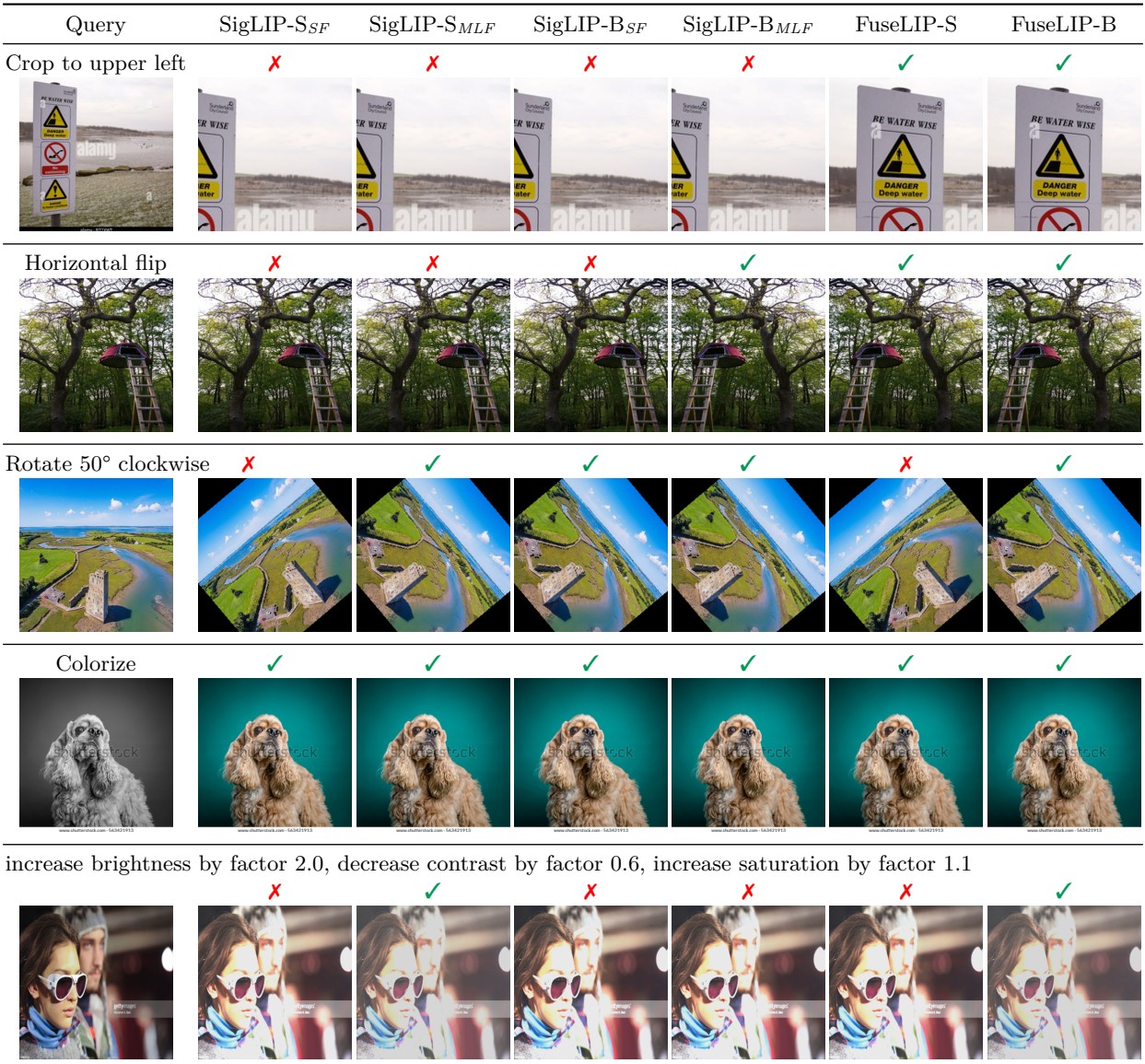

Figure 5: **CC3M-TGIT evaluation examples.** We illustrate the tasks in CC3M-TGIT, together with the prediction of the embedding models.

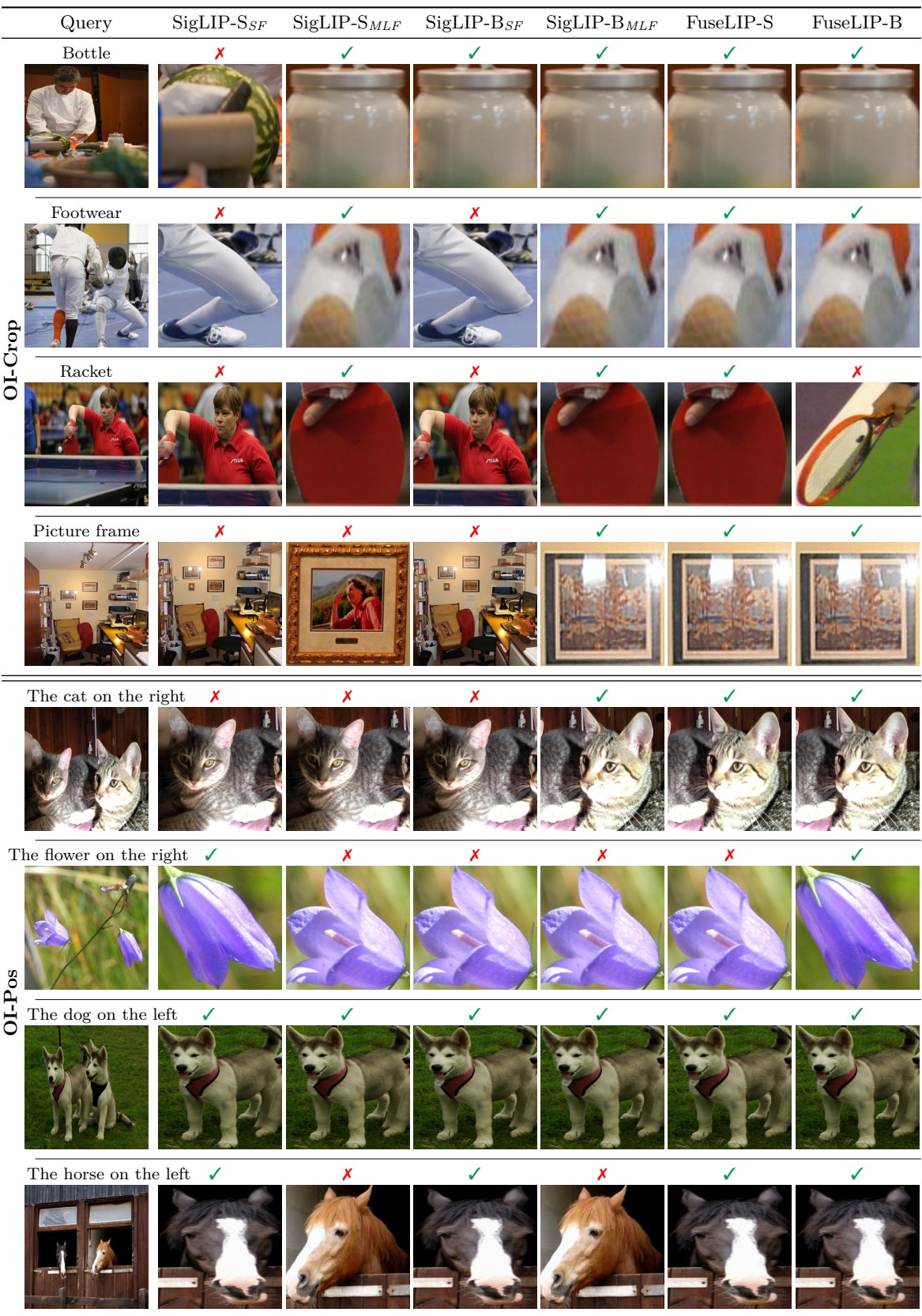

Figure 6: **OI-Crop and OI-Pos.** We show images retrieved on the OI-Crop and OI-Pos tasks by models trained on CC12M+MM.

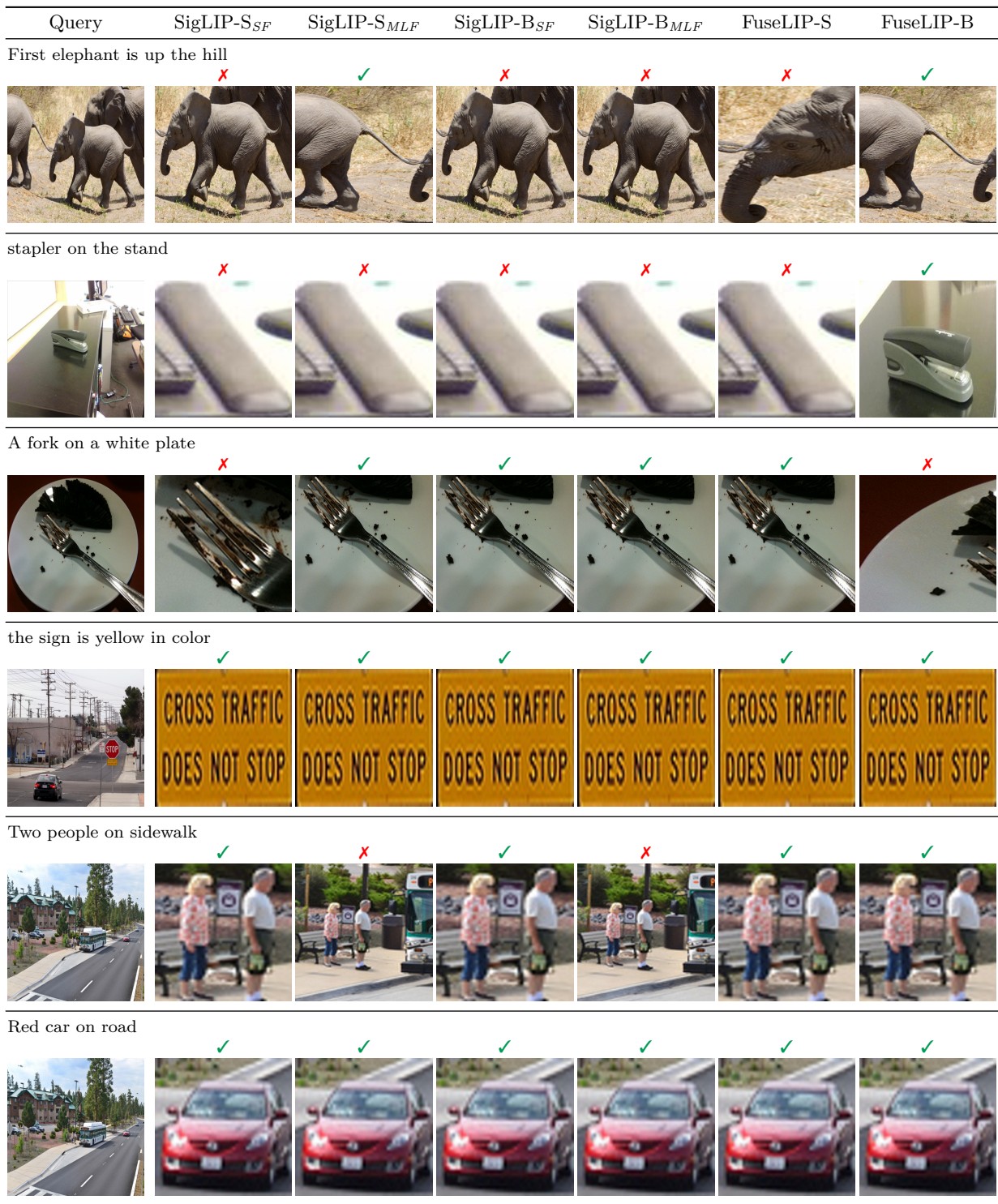

Figure 7: **VG-Crop.** We show images retrieved on the VG-Crop task by models trained on CC12M+MM.

Table 20: **MMEB breakdown** for models trained on CC3M+MM.

| | | SigLIP-S$_{SF}$ | SigLIP-S$_{MLF}$ | SigLIP-B$_{SF}$ | SigLIP-B$_{MLF}$ | FuseLIP-S | FuseLIP-B |
|---|---|---|---|---|---|---|---|
| Classification | ImageNet-1K | 10.7 | 11.1 | 14.2 | 13.8 | 12.3 | 17.4 |
| | CIFAR-100 | 16.7 | 16.2 | 18.1 | 20.4 | 13.8 | 26.1 |
| | N24News | 27.9 | 23.4 | 28.7 | 21.2 | 24.2 | 25.5 |
| | HatefulMemes | 51.3 | 51.0 | 51.8 | 51.0 | 50.6 | 52.2 |
| | VOC2007 | 70.2 | 40.4 | 69.5 | 39.9 | 43.0 | 43.6 |
| | SUN397 | 24.0 | 25.5 | 25.4 | 29.3 | 28.1 | 38.5 |
| | Place365 | 18.8 | 15.0 | 17.6 | 17.8 | 17.2 | 23.4 |
| | ImageNet-A | 0.6 | 0.9 | 0.8 | 1.3 | 0.4 | 0.6 |
| | ImageNet-R | 11.5 | 8.7 | 12.7 | 12.7 | 6.8 | 20.0 |
| | ObjectNet | 4.2 | 5.0 | 4.8 | 6.1 | 6.1 | 7.7 |
| | Country211 | 0.4 | 1.2 | 0.4 | 1.1 | 1.2 | 1.4 |
| | **Mean** | 21.5 | 18.0 | 22.2 | 19.5 | 18.5 | 23.3 |
| VQA | OK-VQA | 20.8 | 19.8 | 20.7 | 21.8 | 23.6 | 26.3 |
| | A-OKVQA | 18.1 | 17.8 | 19.0 | 19.9 | 18.6 | 22.8 |
| | DocVQA | 2.6 | 3.4 | 2.8 | 3.5 | 3.7 | 4.5 |
| | InfographicsVQA | 2.0 | 3.6 | 2.7 | 2.7 | 3.6 | 4.1 |
| | ChartQA | 5.4 | 3.2 | 4.4 | 2.6 | 4.0 | 4.2 |
| | Visual7W | 22.3 | 23.8 | 25.0 | 25.3 | 24.6 | 29.7 |
| | ScienceQA | 3.0 | 4.2 | 4.6 | 4.4 | 4.2 | 4.4 |
| | VizWiz | 3.7 | 4.2 | 3.8 | 3.9 | 5.2 | 6.0 |
| | GQA | 42.5 | 55.0 | 46.6 | 57.3 | 65.3 | 66.4 |
| | TextVQA | 6.7 | 6.5 | 6.1 | 6.3 | 6.1 | 6.7 |
| | **Mean** | 12.7 | 14.2 | 13.6 | 14.8 | 15.9 | 17.5 |
| Retrieval | VisDial | 6.4 | 6.0 | 7.1 | 7.2 | 1.9 | 0.9 |
| | CIRR | 13.4 | 13.8 | 13.2 | 14.5 | 10.2 | 15.3 |
| | VisualNews-t2i | 9.3 | 8.6 | 10.9 | 8.5 | 7.2 | 9.1 |
| | VisualNews-i2t | 10.4 | 9.2 | 11.7 | 10.6 | 7.7 | 11.7 |
| | MSCOCO-t2i | 23.3 | 25.5 | 24.5 | 29.0 | 23.1 | 32.3 |
| | MSCOCO-i2t | 20.1 | 21.4 | 20.8 | 24.7 | 21.3 | 31.0 |
| | NIGHTS | 44.2 | 42.7 | 42.9 | 45.6 | 38.2 | 46.0 |
| | WebQA | 8.4 | 10.0 | 8.5 | 10.0 | 13.5 | 15.6 |
| | FashionIQ | 5.7 | 4.6 | 7.1 | 4.3 | 2.2 | 5.3 |
| | Wiki-SS-NQ | 1.0 | 0.9 | 0.9 | 1.1 | 0.9 | 1.1 |
| | OVEN | 7.3 | 5.2 | 6.4 | 6.9 | 4.5 | 6.4 |
| | EDIS | 6.7 | 4.7 | 6.6 | 3.9 | 4.1 | 5.9 |
| | **Mean** | 13.0 | 12.7 | 13.4 | 13.9 | 11.2 | 15.0 |
| Grounding | MSCOCO | 69.7 | 73.9 | 72.1 | 73.9 | 64.4 | 78.9 |
| | RefCOCO | 81.8 | 82.8 | 83.5 | 85.5 | 76.4 | 91.3 |
| | RefCOCO-Matching | 60.1 | 63.1 | 61.8 | 63.1 | 65.0 | 68.8 |
| | Visual7W-Pointing | 87.7 | 77.0 | 91.3 | 85.0 | 77.3 | 90.8 |
| | **Mean** | 74.8 | 74.2 | 77.2 | 76.9 | 70.8 | 82.4 |

Table 21: **MMEB breakdown** for models trained on CC12M+MM.

| | | SigLIP-S$_{SF}$ | SigLIP-S$_{MLF}$ | SigLIP-B$_{SF}$ | SigLIP-B$_{MLF}$ | FuseLIP-S | FuseLIP-B |
|---|---|---|---|---|---|---|---|
| Classification | ImageNet-1K | 25.8 | 29.6 | 28.7 | 29.7 | 25.0 | 31.2 |
| | CIFAR-100 | 23.7 | 29.1 | 28.0 | 29.7 | 21.1 | 35.7 |
| | N24News | 34.9 | 15.7 | 35.2 | 27.3 | 24.8 | 29.1 |
| | HatefulMemes | 50.0 | 51.0 | 48.8 | 52.3 | 48.6 | 50.7 |
| | VOC2007 | 78.6 | 56.6 | 77.9 | 60.9 | 52.7 | 55.8 |
| | SUN397 | 43.6 | 46.4 | 45.1 | 49.3 | 41.4 | 50.5 |
| | Place365 | 30.5 | 29.3 | 28.3 | 26.6 | 28.4 | 32.2 |
| | ImageNet-A | 1.2 | 1.0 | 1.7 | 2.0 | 0.9 | 1.3 |
| | ImageNet-R | 31.7 | 38.7 | 37.4 | 36.4 | 21.5 | 38.5 |
| | ObjectNet | 10.4 | 11.7 | 11.7 | 14.3 | 9.2 | 13.5 |
| | Country211 | 4.3 | 4.5 | 3.9 | 4.4 | 3.6 | 4.6 |
| | **Mean** | 30.4 | 28.5 | 31.5 | 30.3 | 25.2 | 31.2 |
| VQA | OK-VQA | 28.7 | 29.1 | 30.3 | 28.0 | 28.8 | 30.9 |
| | A-OKVQA | 26.3 | 24.2 | 26.0 | 25.5 | 24.4 | 28.2 |
| | DocVQA | 3.0 | 3.5 | 3.1 | 3.5 | 5.9 | 4.2 |
| | InfographicsVQA | 2.8 | 3.4 | 3.9 | 4.2 | 4.4 | 4.9 |
| | ChartQA | 1.7 | 2.7 | 2.5 | 1.9 | 3.0 | 2.9 |
| | Visual7W | 30.0 | 30.2 | 29.5 | 32.1 | 31.3 | 35.3 |
| | ScienceQA | 3.4 | 4.8 | 4.2 | 2.9 | 4.7 | 5.4 |
| | VizWiz | 5.3 | 6.9 | 6.0 | 6.9 | 6.6 | 8.7 |
| | GQA | 52.3 | 55.0 | 54.2 | 54.7 | 63.9 | 68.0 |
| | TextVQA | 8.2 | 8.8 | 9.9 | 8.4 | 9.2 | 9.2 |
| | **Mean** | 16.2 | 16.9 | 17.0 | 16.8 | 18.2 | 19.8 |
| Retrieval | VisDial | 9.2 | 7.4 | 11.6 | 6.4 | 10.3 | 13.9 |
| | CIRR | 22.6 | 22.5 | 22.0 | 23.5 | 15.1 | 22.5 |
| | VisualNews-t2i | 26.2 | 25.5 | 28.2 | 27.9 | 21.5 | 27.5 |
| | VisualNews-i2t | 25.0 | 24.4 | 26.5 | 26.7 | 21.8 | 27.0 |
| | MSCOCO-t2i | 43.4 | 42.3 | 44.3 | 41.4 | 36.5 | 46.2 |
| | MSCOCO-i2t | 38.3 | 40.9 | 41.1 | 40.7 | 36.8 | 46.2 |
| | NIGHTS | 56.8 | 58.9 | 47.6 | 49.7 | 49.1 | 57.7 |
| | WebQA | 21.3 | 22.0 | 22.7 | 25.0 | 22.8 | 24.6 |
| | FashionIQ | 9.0 | 4.5 | 10.6 | 5.8 | 4.8 | 8.2 |
| | Wiki-SS-NQ | 2.9 | 2.9 | 4.0 | 3.9 | 1.4 | 3.2 |
| | OVEN | 12.0 | 13.2 | 9.8 | 13.6 | 7.9 | 16.8 |
| | EDIS | 18.9 | 14.2 | 17.0 | 13.2 | 13.6 | 21.1 |
| | **Mean** | 23.8 | 23.2 | 23.8 | 23.2 | 20.1 | 26.2 |
| Grounding | MSCOCO | 72.3 | 73.7 | 73.5 | 78.0 | 71.8 | 80.9 |
| | RefCOCO | 78.5 | 81.5 | 78.6 | 83.4 | 82.6 | 91.7 |
| | RefCOCO-Matching | 59.3 | 62.2 | 57.2 | 61.9 | 71.5 | 71.5 |
| | Visual7W-Pointing | 86.5 | 73.3 | 81.4 | 70.3 | 74.8 | 85.0 |
| | **Mean** | 74.2 | 72.7 | 72.7 | 73.4 | 75.2 | 82.3 |

