# OpenReview forum: "FuseLIP: Multimodal Embeddings via Early Fusion of Discrete Tokens"
_TMLR — Under review for TMLR_

### Review · Reviewer_MAoJ · 2026-05-26

**Summary Of Contributions:**

This paper proposes **FuseLIP**, a multimodal embedding architecture that replaces the conventional dual-encoder plus fusion pipeline, with a single transformer operating over a unified vocabulary of text tokens and discrete image tokens.

Image and text tokens are concatenated immediately after tokenization, enabling "early fusion" of modalities throughout the encoder rather than only in late layers. Training uses two losses at the same time. The first is a contrastive loss similar to SigLIP[1]'s sigmoid version that pulls matching image–text pairs together in the embedding space and pushes non-matching pairs apart. The second is a masked-token prediction loss: some input tokens are hidden, and the model has to guess them, like fill-in-the-blank, but for both image tokens and text tokens at once. Because images and text are already turned into discrete tokens drawn from one shared vocabulary, the same prediction head can handle both modalities. FLAVA[2], by contrast, needs a separate tokenizer just for masking images and a separate prediction head for each modality.

Key strengths:
* Genuinely controlled comparison: baselines are trained from scratch on the same data and with the same hard-negative recipe, which makes the late-vs-early-fusion claim testable. Most prior work in this area mixes architectural changes with massive data-scale differences.
* The TGIT result is striking and well-explained. Authors reveal that late-fusion models cannot solve text-guided crop/rotation/flip retrieval at training (~50–60% for SF on crop, ~55–58% on flip), whereas FuseLIP solves them near perfectly. The hypothesis that deep semantic embeddings discard the low-level visual cues these tasks need, is plausible and consistent with the SF vs MLF gap pattern.
* The MMM-loss integration is architecturally clean, and the ablation (Table 6) shows a sizeable benefit, especially for FuseLIP-B. The combination of contrastive + MMM in one forward pass is a real engineering simplification over FLAVA.
* Thorough appendix that covers per-MMEB-subtask breakdowns, qualitative retrievals (Figs. 4–6), memory and runtime tables, loss-only ablation, and unimodal-only training, that all support the claims.
* Honest reporting of weaknesses: higher inference cost, in-domain nature of TGIT/VG-Crop training, parity on standard unimodal tasks, etc.

Key weaknesses:
* The dramatic TGIT and VG-Crop numbers in Table 3 are in-distribution, since those datasets appear in training. This substantially deflates the headline narrative. The genuinely zero-shot gains on OI-Pos, OI-Crop, and MMEB Grounding/VQA are real but more modest.
* The claim that early fusion is the active ingredient is partly confounded by tokenizer choice. Table 12 shows that swapping TiTok-S for TiTok-B with a fixed small transformer gives nearly as much improvement as the full FuseLIP-B, so the design does not fully separate "early fusion of discrete tokens" from "high-quality compressed image tokens."
* No reported variance or seeds. Several gaps on MMEB sub-areas are within a few percentage points and plausibly within run-to-run noise.
* The hard-negative recipe is meticulously hand-tuned per training source (App. A.3). Removing it collapses TGIT performance from 94.3% to 13.6% (Table 6), which raises the question of how much of the win is the architecture versus the curriculum.
* The comparison to pretrained models in Table 5 is appropriately hedged, but VLM2Vec[3] beats all FuseLIP variants on essentially every MMEB category, despite the authors' claim around FuseLIP wins on "challenging multimodal tasks", since those tasks are the authors' own.

[1] Zhai, Xiaohua, et al. "Sigmoid loss for language image pre-training." Proceedings of the IEEE/CVF international conference on computer vision. 2023.
[2] Singh, Amanpreet, et al. "Flava: A foundational language and vision alignment model." Proceedings of the IEEE/CVF conference on computer vision and pattern recognition. 2022.
[3] Jiang, Ziyan, et al. "Vlm2vec: Training vision-language models for massive multimodal embedding tasks." ICLR 2025.

**Audience:**

Yes

**Audience Explanation:**

Multimodal embeddings are an active area with practical relevance to retrieval, RAG, agent context, and as substrates for VLM training. The community has largely converged on late-fusion CLIP variants, and a controlled demonstration that a single-encoder, early-fusion alternative is feasible and qualitatively better when visual structure matters more than caption-level semantics is a non-trivial contribution.

Three findings will plausibly be reused by TMLR readers:

* The TGIT-style diagnostic is a useful probe. Score-fused CLIP failing to identify a rotated or cropped version of its own input given a textual instruction is a clean failure mode of late fusion, relevant to anyone building controllable, instruction-following retrieval.
* The architectural recipe presented in this paper is small and simple enough to serve as a starting point for follow-up work, including the scaling studies this paper could not run.
* The training-data construction patterns (TGIT, LLM-generated VQA from captions, etc.) are useful artifacts by their own.

The negative findings are also informative: at academic budget, fine-tuned FuseLIP does not dominate the much larger pretrained VLM2Vec on standard MMEB tasks (Table 5). This honest comparison is worth reporting.

**Broader Impact Concerns:**

The paper has no Broader Impact Statement.

Given the methodological nature of the contribution, this is not disqualifying, but a brief statement should be added covering:
1. inherited biases from CC3M/CC12M and OpenImages
2. the use of Llama-3.1-8B-Instruct to generate CC3M-VQA from captions alone, which can encode LLM priors and caption biases;
3. the OI-Pos capability, which has obvious benign uses but is also relevant to identification or surveillance contexts.

One paragraph is sufficient.

**Claims And Evidence:**

Yes

**Claims Explanation:**

The headline claims hold up. The architecture comparison is fair: baselines are trained from scratch on the same data with the same hard-negative recipe, and FuseLIP-B comes out ahead on most task categories in Table 3. The MMM-loss and hard-negative ablations in Table 6 show clear, directionally consistent effects across both model sizes, and the Table 4 breakdown gives a satisfying explanation for why late fusion fails on Crop and Flip while solving the semantically rich transformations.

Two caveats keep this from being unqualified. First, the most dramatic numbers (TGIT, VG-Crop) are in-distribution, and the paper occasionally leans on them in framing without flagging this. Second, Table 12 only partially separates the contribution of "early fusion" from "good discrete tokenizer plus matched training". The tokenizer alone moves the needle substantially.

**Requested Changes:**

Critical changes that I recommend:
1. **Disentangle "early fusion" from "tokenizer quality."** Table 12 shows that pairing TiTok-B with a small transformer outperforms TiTok-S with a base transformer on most metrics. This suggests a meaningful share of FuseLIP's gains comes from tokenizer fidelity rather than the single-encoder design itself. To separate these two factors, please add a baseline that feeds TiTok-B tokens into a late-fusion setup. For example, one encoder for the discrete image tokens, a separate text encoder, then score- or MagicLens-style fusion. Even a single row would tell us whether the win comes from processing discrete tokens through one encoder or simply from using high-quality discrete tokens at all. At minimum, please acknowledge this confound explicitly in Sec. 5.2 or Sec. 6.
2. **Flag the in-distribution status of TGIT and VG-Crop.** The ⋆ marker in Table 3 is technically correct, but the abstract and Sec. 5.2 highlight TGIT (94.3%) as a flagship result without noting that TGIT is also a training source. Please add an explicit caveat in "Why early fusion helps to understand text-guided transformations" clarifying that this number measures learnability, not generalization. To make a generalization claim, consider holding out one transformation family (e.g., train without flip, test on flip).
3. **Report variance.** Please run at least one configuration with 3 seeds and report mean±std for Tables 3, 5, and 6 — FuseLIP-S on CC3M+MM is cheap enough to make this practical. Several MMEB sub-area gaps in Table 3 are in the 1–3% range, and the SugarCrepe gaps in Table 10 are often 1–2%; without variance estimates, these differences cannot be interpreted.

Optional changes that would strengthen the paper
1. Hard negative sensitivity. Table 6 removes hard negatives entirely. For example, keeping hard negatives only for TGIT, or only for VG-Crop would show which task families actually drive the dependence and help readers building on this recipe decide where to invest curriculum effort.
2. Move qualitative evidence into the main text. One row of Fig. 4 (e.g., crop-to-upper-left) could be more persuasive than Table 4 alone, since late-fusion baselines visibly return the original orientation. Promoting one example into Sec. 5.2 would strengthen the narrative.
3. Inference-cost discussion. Table 7 shows FuseLIP-B is roughly 35% slower at inference than SigLIP-BMLF, and the "we expect this to shrink at scale" framing in Sec. 6 is thin. A breakdown of tokenizer versus encoder time would help, along with a note that for retrieval indexing, image embeddings can be precomputed, so the tokenizer cost amortizes.
4. Reproducibility statement. The paper does not state whether code, the generated datasets (CC3M/12M-TGIT, CC3M-VQA, OI-Crop, OI-Pos), or model checkpoints will be released.
5. Trainable-parameter framing (Table 1). The "fewer trainable parameters" claim is fair as a training-cost statement, but slightly misleading at inference: the tokenizer is frozen, not absent. Please tighten the wording in Sec. 5 accordingly.

---

> ### Author Response · Authors · 2026-06-22
> **Rebuttal [1/2]**
>
> We thank Reviewer MAoJ for their detailed and constructive feedback. We appreciate that they highlight the **"genuinely controlled comparison"** as a key strength, note that the **"TGIT result is striking and well-explained"**, and commend the **"architecturally clean"** MMM-loss integration. We have revised the manuscript according to the reviewers' comments, with **changes colored in teal**. In the following, we address the concerns individually.
>
>  &nbsp;
>
> **Weaknesses**
>
>  &nbsp;
>
> > *W1, W2, W3*
>
> Addressed in detail under "Critical changes" below.
>
>  &nbsp;
>
> > *W4. The hard-negative recipe is meticulously hand-tuned per training source (App. A.3). Removing it collapses TGIT performance from 94.3% to 13.6% (Table 6), which raises the question of how much of the win is the architecture versus the curriculum.*
>
> We agree that hard negatives are crucial for learning TGIT, and we do not claim that FuseLIP would solve these tasks without them. Rather, we view the hard-negative sampling strategy as an important part of our training recipe and as one of the contributions of the paper. Table 6 confirms that these negatives provide an essential training signal for several multimodal tasks. However, **this does not explain the relative advantage of FuseLIP over late-fusion baselines, since the same hard-negative strategy is used for all models.** Under this controlled comparison, FuseLIP still substantially outperforms the late-fusion baselines on TGIT.
>
>  &nbsp;
>
> > *W5. The comparison to pretrained models in Table 5 is appropriately hedged, but VLM2Vec beats all FuseLIP variants on essentially every MMEB category, despite the authors' claim around FuseLIP wins on "challenging multimodal tasks", since those tasks are the authors' own.*
>
> We agree that FuseLIP does not outperform much larger or substantially longer-pretrained state-of-the-art models such as VLM2Vec on MMEB. The controlled comparison in our paper is instead between FuseLIP and late-fusion baselines trained from scratch on the same data, with the same hard-negative strategy and comparable model scale. In this setting, **FuseLIP-B outperforms the strongest late-fusion baselines on most MMEB categories.** The comparison to VLM2Vec serves as context rather than a controlled architectural comparison: VLM2Vec is based on a 4.15B-parameter VLM, over 25 times larger than FuseLIP-B, and has undergone substantially longer pretraining. Moreover, we have revised the manuscript to specify that the challenging multimodal tasks where FuseLIP outperforms VLM2Vec are created by us.
>
>  &nbsp;
>
> **Critical changes**
>
>  &nbsp;
>
> > *C1. Disentangle "early fusion" from "tokenizer quality." … please add a baseline that feeds TiTok-B tokens into a late-fusion setup. …*
>
> We added a new late-fusion baseline to disentangle tokenizer fidelity from the early-fusion architecture. In this baseline, the image branch of SigLIP-B-MLF receives discrete image tokens from TiTok-B, while the text branch remains separate and fusion is performed only after encoding. This directly tests whether high-quality discrete image tokens alone explain the gains. We train on CC3M+MM. As shown in Table A below, **this baseline performs worse than the standard SigLIP-B-MLF baseline on most metrics** (e.g. Grounding drops from 76.9 to 68.2 and OI-Crop from 68.4 to 58.8). These results indicate that FuseLIP's gains are not primarily explained by tokenizer fidelity, but by jointly processing image and text tokens in an early-fusion encoder. We have added this experiment to the appendix of the revised manuscript.
>
> *Table A: Late fusion baseline with discrete image tokens.*
>
>
> | Model                | Classification | VQA      | Retrieval | Grounding | ImageNet | VG-Crop  | OI-Crop  | OI-Pos   | TGIT     |
> | -------------------- | -------------- | -------- | --------- | --------- | -------- | -------- | -------- | -------- | -------- |
> | SigLIP-B_MLF         | 19.5           | 14.8     | 13.9      | 76.9      | 12.2     | 55.4     | **68.4** | 47.4     | 69.4     |
> | TiTok-B-SigLIP-B_MLF | 18.6           | 14.2     | 11.8      | 68.2      | 12.1     | 51.1     | 58.8     | 47.3     | 69.3     |
> | FuseLIP-B            | **23.3**       | **17.5** | **15.0**  | **82.4**  | **18.1** | **55.8** | 68.1     | **70.8** | **94.3** |

---

> > ### Author Response · Authors · 2026-06-22
> > **Rebuttal [2/2]**
> >
> > > *C2. Flag the in-distribution status of TGIT and VG-Crop. The ⋆ marker in Table 3 is technically correct, but the abstract and Sec. 5.2 highlight TGIT (94.3%) as a flagship result without noting that TGIT is also a training source. Please add an explicit caveat … clarifying that this number measures learnability, not generalization. …*
> >
> > We agree that TGIT and VG-Crop task families are part of the training mixture, while the evaluation uses held-out samples. So these evaluations should be interpreted as measuring learnability of these tasks rather than zero-shot generalization to unseen task types. We have clarified this in the revised manuscript in the paragraph "Why early fusion helps to understand text-guided transformations".
> >
> > At the same time, we would like to emphasize that the zero-shot gains are not limited to small effects. **In particular, OI-Pos is not part of the training mixture and shows a large improvement**: trained on CC3M+MM, FuseLIP-B reaches 70.8 compared to 47.4 for the strongest late-fusion baseline, and trained on CC12M+MM it reaches 68.9 compared to 48.9. Thus, OI-Pos provides an indication that early fusion improves zero-shot multimodal embedding.
> >
> >  &nbsp;
> >
> > > *C3. Report variance. Please run at least one configuration with 3 seeds and report mean±std …*
> >
> > We conduct a multi-seed experiment in Table B below, where we train FuseLIP-B and the strongest late-fusion baseline SigLIP-B_MLF on CC3M+MM with three different random seeds and report mean and standard deviation. **The results show that the main gains are stable across seeds: FuseLIP substantially improves over the baseline on most tasks, with differences well above the observed run-to-run variation.** At the same time, VG-Crop and OI-Crop are comparable within variance. We have added this experiment to the appendix of the revised manuscript.
> >
> > *Table B: Variance over training random seeds.*
> >
> >
> > | Model        | Classification | VQA            | Retrieval      | Grounding      | ImageNet       | VG-Crop        | OI-Crop        | OI-Pos         | TGIT           |
> > | ------------ | -------------- | -------------- | -------------- | -------------- | -------------- | -------------- | -------------- | -------------- | -------------- |
> > | SigLIP-B-MLF | 20.2 ± 0.8     | 14.3 ± 0.4     | 14.0 ± 0.2     | 75.7 ± 1.0     | 12.3 ± 0.4     | **55.9** ± 0.8 | **67.8** ± 0.8 | 47.5 ± 0.2     | 70.9 ± 1.7     |
> > | FuseLIP-B    | **23.0** ± 0.5 | **17.2** ± 0.5 | **15.0** ± 0.8 | **82.1** ± 0.6 | **17.2** ± 1.5 | **55.6** ± 1.9 | **67.4** ± 1.2 | **69.0** ± 1.5 | **91.2** ± 5.5 |
> >
> >
> >  &nbsp;
> >
> > **Optional**
> >
> >  &nbsp;
> >
> > > *O1. Hard negative sensitivity. Table 6 removes hard negatives entirely. For example, keeping hard negatives only for TGIT, or only for VG-Crop would show which task families actually drive the dependence …*
> >
> > We agree that a per-dataset ablation would provide an interesting analysis, but it is beyond our current scope and we leave it to future work. Our full hard-negative ablation in Table 6 directly supports our main claim that hard negatives are a crucial component of the proposed training recipe.
> >
> >  &nbsp;
> >
> > > *O2. Move qualitative evidence into the main text. One row of Fig. 4 (e.g., crop-to-upper-left) could be more persuasive than Table 4 alone …*
> >
> > We now show one row of the TGIT Figure in the main part of the revised manuscript (Figure 3), in order to better demonstrate the nature of the TGIT tasks and how models solve them.
> >
> >  &nbsp;
> >
> > > *O3. Inference-cost discussion …*
> >
> > We have added a note on the amortization of tokenizer cost for retrieval indexing in Section A.2 in the revised manuscript.
> >
> >  &nbsp;
> >
> > > *O4. Reproducibility statement …*
> >
> > The source code, generated datasets, and trained models will be **publicly released.**
> >
> >  &nbsp;
> >
> > > *O5. Trainable-parameter framing (Table 1). The "fewer trainable parameters" claim is fair as a training-cost statement, but slightly misleading at inference: the tokenizer is frozen, not absent. Please tighten the wording in Sec. 5 accordingly.*
> >
> > We agree that the tokenizer influences evaluation runtime and have added a clarifying note in Sec. 5.
> >
> >  &nbsp;
> >
> > > *O6. The paper has no Broader Impact Statement. …*
> >
> > We have added a broader impact statement to the revised manuscript, discussing the points raised by the reviewer.
> >
> >  &nbsp;
> >
> > We thank the reviewer again for their constructive comments. We are happy to answer any further questions.

---

### Review · Reviewer_HYHX · 2026-06-05

**Summary Of Contributions:**

The paper introduces a novel architecture and design for generating unified embeddings from multimodal inputs. In contrast to baseline models such as CLIP, which encode different modalities separately and fuse them at a later stage, this work represents different modalities through tokenization within a unified library, enabling fusion at an early stage.

**Strengths:**

* The paper is well written and provides extensive experiments and ablation studies. The experimental details are sufficiently clear to support reproducibility, and the analysis can be insightful for future work in this area.
* Beyond its technical novelty, the paper introduces a benchmark with new datasets involving combinations of text and image as input and/or output, which may be useful for future research.
* The paper proposes a multimodal framework that can be easily adapted to different input and output modalities, including text, image, and text+image. This addresses the adaptation challenges faced by previous methods when handling combined modality formats.
* FuseLiP demonstrates superior performance on most tasks in both zero-shot and fine-tuning scenarios.

**Weaknesses:**

* The section discussing the use of hard negative examples is somewhat vague. Providing more explanation about how these examples are selected and used would improve clarity.
* Some validation scenarios are not fully clear, such as the ImageNet evaluation. Additional details about the experimental setup would be helpful.
* The FuseLiP-S model performs worse than the baseline in some scenarios, but the paper provides limited explanation or discussion of these cases. A more detailed analysis of these failure cases would strengthen the paper.

**Audience:**

Yes

**Audience Explanation:**

I found the paper very interesting, especially given the rapidly growing use of multimodal scenarios and the fact that many current backbones are built upon these types of models. I believe many readers will find this paper engaging and valuable, as they can learn a great deal from its extensive ablation studies and experimental analyses.

**Claims And Evidence:**

Yes

**Claims Explanation:**

The paper’s main contribution claims regarding its performance in both multimodal and unimodal settings are well supported by the results presented in the tables. The authors’ claim of introducing new evaluation tasks for multimodal scenarios is also clearly supported in the paper through detailed descriptions and strong explanations.

In addition, the paper argues that early fusion is more effective than late fusion models. This claim is supported both conceptually and empirically. The authors explain that while semantic features can be obtained at later stages, some scenarios require the model to select or prioritize features based on information from other modalities. The reported results further support this argument by showing the effectiveness of early fusion compared with late fusion baselines.

Moreover, regarding the use of hard negative examples and the masking loss, the authors provide ablation studies on their own architecture, in addition to comparisons with baseline models. These experiments help demonstrate the effectiveness of these components and support their contribution to the overall performance.

**Requested Changes:**

**Critical:**

1. One important question regarding this paper is related to tokenization. Since both image and text inputs are tokenized, what happens in scenarios where both text and image are used as input, but the number of tokens in one modality is much larger than in the other? Could this introduce a bias toward one modality? In conventional methods, the embedding dimensions are usually within a similar range, but here the number of tokens can vary across different tasks. Providing an explanation regarding this issue would be very helpful.

2. In the “Early Fusion” paragraph in Section 3.1, you mention the use of positional embeddings. Are the positional embeddings applied to each modality separately or jointly after concatenation? This is important because positional information has meaning within each modality, but it may also be affected by the ordering of modalities, for example placing the image first and the text afterward. Does the order of modalities matter in your framework? Further explanation on this point would be helpful.

3. For tasks where the model is expected to output text, does the model select from a set of predefined choices, or does it generate the output from scratch? If the output is generated from scratch, how do you handle multi-token outputs?

4. In Table 3, for some scenarios such as classification, retrieval, and grounding, some of the S-scale baselines perform better than FuseLiP-S. Why is this the case? This pattern appears less frequently for FuseLiP-B. Providing an explanation for this behavior would be very informative.

**Minor:**

1. Providing more information about the hard negative examples would make the paper easier to read and understand.

2. In the encoder paragraph of Section 3.1, the sentence “The final embedding corresponds to the `<eot>` token’s representation” is confusing. Could you clarify what this means?

3. The descriptions of some tasks, such as ImageNet, are vague. Adding more details about these evaluation settings would be helpful.

4. On page 9, the paper states: “This is further supported by the performance gap between the late fusion baselines: SigLIPMLF, which uses a learnable deep fusion network, outperforms SigLIPSF, which simply sums the unimodal embeddings.” This statement should be made more carefully, since even in Table 4, the results differ substantially between SF and MLF in only two out of four settings, while in the other two settings, their performances are relatively close in S-scale.

---

> ### Author Response · Authors · 2026-06-22
> **Rebuttal [1/2]**
>
> We thank Reviewer HYHX for their detailed and constructive feedback. We appreciate that they find the paper **"well written"** with **"extensive experiments and ablation studies"**, note that the experimental details are **"sufficiently clear to support reproducibility"**, and highlight that FuseLIP **"demonstrates superior performance on most tasks in both zero-shot and fine-tuning scenarios"**. We are also glad they consider the new benchmark and datasets useful for future research. We have revised the manuscript according to the reviewers' comments, with **changes colored in orange**. In the following, we address the concerns individually.
>
> &nbsp;
>
> **Critical**
>
> &nbsp;
>
> > *C1. Since both image and text inputs are tokenized, what happens in scenarios where both text and image are used as input, but the number of tokens in one modality is much larger than in the other? Could this introduce a bias toward one modality? …*
>
> In FuseLIP, the number of tokens affects the sequence length, but not the dimensionality of the resulting embedding: every token is mapped to a d-dimensional vector, and the final output is a single d-dimensional representation obtained from the final representation of the `<eot>` ("end of text") token. A token-count imbalance could in principle influence the attention distribution. However, **empirically, we do not observe a collapse towards the modality with more tokens**: tasks such as VQA, visual grounding, OI-Pos, and TGIT require using both modalities, and FuseLIP outperforms or matches baselines on these tasks. In practice, image inputs always contribute exactly 128 tokens (out of a total context length of 180), while text queries in image+text tasks typically occupy 10–25 tokens (well within the remaining 49-token budget) so usually no truncation of meaningful text content occurs.
>
> &nbsp;
>
> > *C2. In the "Early Fusion" paragraph in Section 3.1, you mention the use of positional embeddings. Are the positional embeddings applied to each modality separately or jointly after concatenation? …*
>
> In FuseLIP, positional embeddings are applied jointly after concatenation. Image discrete tokens (N = 128) are prepended to text tokens to form a single flat sequence, and a single shared learnable positional embedding is added to this full sequence. The ordering is fixed throughout training and inference (image always first), so the model learns a consistent positional code for each modality's slot. **Crucially, FuseLIP uses bidirectional attention, so all tokens attend to all others regardless of position, and there is no asymmetric information flow between modalities.** We have specified the positional embedding in Section 3.1 in the revised manuscript.
>
> &nbsp;
>
> > *C3. For tasks where the model is expected to output text, does the model select from a set of predefined choices, or does it generate the output from scratch? …*
>
> The text is selected from a set of predefined choices per task, in line with the embedding model literature and benchmarks. We have specified this in Section 5.1 in the revised manuscript.
>
> &nbsp;
>
> > *C4. In Table 3, for some scenarios such as classification, retrieval, and grounding, some of the S-scale baselines perform better than FuseLiP-S. Why is this the case? …*
>
> We believe this behavior is mainly due to capacity allocation. **FuseLIP-S has a similar *total* parameter count as the S-scale baselines, but substantially fewer *trainable* parameters because the image tokenizer is frozen** (see Table 1). This can make the small-scale model less competitive on some tasks. The pattern becomes less frequent for FuseLIP-B because scaling improves both relevant components: it uses a stronger TiTok-B image tokenizer and a larger transformer encoder. Our tokenizer/transformer ablation in Table 13 further supports this: switching from the S to the B tokenizer already gives considerable gains, and scaling the transformer yields further improvements. We have added this discussion in Section 5.2 in the revised manuscript.

---

> > ### Author Response · Authors · 2026-06-22
> > **Rebuttal [2/2]**
> >
> > **Minor**
> >
> > &nbsp;
> >
> > > *M1. Providing more information about the hard negative examples would make the paper easier to read and understand.*
> >
> > We have extended the explanation on our hard negative scheme in Section 4.3.
> >
> > &nbsp;
> >
> > > *M2. In the encoder paragraph of Section 3.1, the sentence "The final embedding corresponds to the `<eot>` token's representation" is confusing. Could you clarify what this means?*
> >
> > After the full token sequence has been processed by the encoder, we take the final-layer representation of the `<eot>` ("end of text") token as the resulting embedding, following the convention of CLIP-style text encoders (Radford et al., 2021). This embedding is then used in the training loss computation and for downstream tasks. Through bidirectional attention, the `<eot>` representation aggregates information from all non-padding tokens. We have clarified this in Section 3.1 in the revised manuscript.
> >
> > &nbsp;
> >
> > > *M3. The descriptions of some tasks, such as ImageNet, are vague. …*
> >
> > We have improved task descriptions, in particular for ImageNet, in Section 5.1 in the revised manuscript.
> >
> > &nbsp;
> >
> > > *M4. On page 9, the paper states: "This is further supported by the performance gap between the late fusion baselines: SigLIP-MLF, which uses a learnable deep fusion network, outperforms SigLIP-SF, which simply sums the unimodal embeddings." This statement should be made more carefully …*
> >
> > We agree that the performance of SigLIP-S_SF and SigLIP-S_MLF is similar on some TGIT subtasks in Table 4. However, the gap is substantial on "crop", "flip", and "jitter". Thus, we believe our claim remains valid: learnable late fusion can improve over simple score fusion on several transformations, but it still does not close the gap to early fusion. We have revised the corresponding paragraph in Section 5.2 to make this statement more precise.
> >
> > &nbsp;
> >
> > We thank the reviewer again for their constructive comments. We are happy to answer any further questions.

---

### Review · Reviewer_X8RX · 2026-06-15

**Summary Of Contributions:**

This paper introduces FuseLIP, a multimodal embedding model that fuses image and text by early fusion of discrete tokens. Images are mapped to 128 discrete tokens by a frozen TiTok tokenizer, concatenated with text tokens over a unified vocabulary and processed by a single bidirectional transformer encoder based on the SigLIP architecture, with the <eot> representation taken as the embedding. This contrasts with late fusion, where each modality is encoded separately and the outputs are merged afterwards.

FuseLIP is trained with two objectives on the same encoder: a SigLIP sigmoid contrastive loss over matched (query, target) pairs and a BERT-style masked multimodal modeling (MMM) loss that the discrete tokenization enables without the auxiliary tokenizers and per-modality heads required by FLAVA. Training relies on hard negatives.

The authors curate new datasets and tasks (CC3M/CC12M-TGIT text-guided transformations, CC3M-VQA, VG-Crop, OI-Crop, OI-Pos) and compare two model sizes against late-fusion SigLIP baselines under matched data and hard negatives. They report that FuseLIP-B is best on nearly all tasks, with significant gains on position/transformation tasks and that both MMM and hard negatives are important.

**Strengths:**

S1. The architecture is simple and clean. Casting both modalities into a shared discrete vocabulary and using a single encoder is the way forward to obtain a unified CLIP-like embedder that also supports a masked-modeling loss without FLAVA's auxiliary tokenizers and heads.

S2. The new evaluation tasks (OI-Pos, OI-Crop, TGIT) deliberately decouple visual structure from semantic content and expose a real, intuitive failure of late fusion. The gains on this set of tasks are large and consistent.

**Weaknesses:**

W1. The central finding that FuseLIP better captures image-text relations and outperforms across nearly all tasks is partially supported by base variant and not necessarily small variant. FuseLIP-S performs poorly on Classification, Retrieval, Grounding and crop tasks VG-Crop and OI-Crop. The tasks where this is clearly evident are OI-Pos, TGIT, VQA and ImageNet so I think it is still a bit mixed results.

W2. The gains mostly concentrate on in-distribution tasks such as OI-Crop/OI-Pos and not on out-of-distribution MMEB where FuseLIP-B lags behind pretrained SigLIP and VLM2Vec by significant margins (10 pp on Classification and 20 pp on ImageNet). The generality of the embedder beyond the proposed tasks is not established similar to W1.

W3. The architecture and the training objective are confounded. The MMM loss, which consistently contributes, can only be applied to FuseLIP and the baselines are contrastive-only by construction. So "early vs. late fusion" partly conflates the fusion strategy with the availability of an auxiliary loss and a clean attribution requires either a FLAVA-style masked baseline or making the no-MMM FuseLIP vs. baseline comparison the primary one.

W4. Table 6 shows that removing hard negatives collapses FuseLIP-B TGIT from 94.3 to 13.6 so it is not clear if the gains are due to hard negatives or fusion.

**Audience:**

Yes

**Audience Explanation:**

The central question of whether it is beneficial and effective to construct multimodal embeddings by early fusion of a shared discrete vocabulary rather than merging separately encoded modalities is interesting and quite relevant given recent advancements in model architectures and training paradigms. The finding that late-fusion embedders are largely blind to visual structure while an early-fusion encoder solves these tasks almost perfectly is an interesting hypothesis with partial evidence over newly proposed tasks on existing benchmarks.

**Broader Impact Concerns:**

No ethical concerns

**Claims And Evidence:**

Yes

**Claims Explanation:**

The claim that early fusion of discrete tokens helps on tasks requiring visual structure (orientation, position, crop identity) is partially supported through the current experimental setup. In Table 6, FuseLIP-B with hard negatives but without MMM still reaches TGIT 88.4 vs 69.4 for the baselines. However, the broader claim of nearly outperforming all the tasks is overstated as the gains are mostly concentrated on the in-distribution tasks and not on out-of-distribution tasks. The early vs late fusion comparison is not fully clean because the baselines cannot use the MMM loss. The claims are therefore supported for the targeted visual-structure tasks, but cannot be generalized at the moment for superior embedding quality.

**Requested Changes:**

The authors must address the concerns highlighted in the Weaknesses section above. Additionally, the authors should include an analysis tying tokenizer fidelity and token budget to downstream accuracy and its implications on the inference cost.

Questions/Suggestions:

1. Please disambiguate the two uses of "masking" (padding/attention masking in Section 3.1 vs. the MMM <MASK> token in Section 3.2) and state explicitly that the encoder is bidirectional because both objectives are non-causal

2. The contrastive loss is computed on the same masked input as the MMM loss. The authors should consider including an ablation that instead computes it on the unmasked input to clarify whether this choice benefits or hurts the contrastive tasks.

---

> ### Author Response · Authors · 2026-06-22
> **Rebuttal [1/2]**
>
> We thank Reviewer X8RX for their detailed and constructive feedback. We appreciate that they describe the architecture as **"simple and clean"**, and highlight the new evaluation tasks as exposing **"a real, intuitive failure of late fusion"** with FuseLIP showing **"large and consistent"** gains. We have revised the manuscript according to the reviewers' comments, with **changes colored in red**. In the following, we address the concerns individually.
>
>  &nbsp;
>
> > *W1. Mixed results for FuseLIP-S.*
>
> We agree that the evidence is strongest for FuseLIP-B, and that FuseLIP-S does not uniformly outperform the late-fusion baselines. We have revised the wording to avoid suggesting that all FuseLIP variants dominate across all tasks. The mixed behavior of FuseLIP-S is likely due to limited capacity and tokenizer fidelity at the smallest scale: it combines a small transformer encoder with the weaker TiTok-S image tokenizer. Notably, while it has a similar total parameter count as the S-sized baselines, it has substantially fewer trainable parameters, since the image tokenizer is frozen. Nevertheless, **even under this smaller trainable-parameter budget, FuseLIP-S already shows clear gains on several tasks requiring image-text interaction, in particular VQA, OI-Pos, and TGIT.** Scaling to FuseLIP-B improves both the tokenizer and encoder capacity and yields the strongest overall results. We therefore clarify in the revised manuscript that the small model shows mixed behavior, while the main evidence for broad performance gains comes from FuseLIP-B.
>
>  &nbsp;
>
> > *W2. The gains mostly concentrate on in-distribution tasks such as OI-Crop/OI-Pos and not on out-of-distribution MMEB where FuseLIP-B lags behind pretrained SigLIP and VLM2Vec …*
>
> We believe there is a misunderstanding regarding the in-distribution status of OI-Crop and OI-Pos. These tasks are constructed from OpenImages and are not part of the training mixture. **Thus, the gains on OI-Crop and OI-Pos are zero-shot with respect to the evaluation dataset**, although OI-Crop is related in task format to VG-Crop. Regarding MMEB, we agree that FuseLIP does not outperform pretrained SigLIP and VLM2Vec on all standard MMEB categories: however, these models are much larger and/or pretrained on substantially larger datasets, thus giving them a significant advantage. The comparison to VLM2Vec serves as context rather than a controlled architectural comparison: **VLM2Vec is based on a 4.15B-parameter VLM, over 25 times larger than FuseLIP-B**, and has undergone substantially longer pretraining. The controlled comparison in our paper is instead between FuseLIP and late-fusion baselines trained from scratch on the same data, with the same hard-negative strategy and comparable model scale. In this setting, **FuseLIP-B outperforms the strongest controlled late-fusion baselines on most MMEB categories**.
>
>  &nbsp;
>
> > *W3. The architecture and the training objective are confounded. The MMM loss, which consistently contributes, can only be applied to FuseLIP and the baselines are contrastive-only by construction …*
>
> We agree that the final FuseLIP model combines the early-fusion architecture with an auxiliary MMM loss, and we have revised Section 5.3 in the manuscript to make this attribution more precise. In particular, we now emphasize the no-MMM ablation in Table 6. This comparison shows that the gains are not solely explained by MMM: **even without the MMM loss, FuseLIP-B remains stronger than the late-fusion baselines on most tasks**, with especially large gains on OI-Pos and TGIT. The MMM loss therefore provides an additional improvement, but is not the sole driver of the observed gains. Notably, this auxiliary objective is naturally supported by the shared discrete-token architecture, whereas a FLAVA-style masked late-fusion baseline would require additional modality-specific tokenizers and prediction heads.
>
>  &nbsp;
>
> > *W4. Table 6 shows that removing hard negatives collapses FuseLIP-B TGIT from 94.3 to 13.6 so it is not clear if the gains are due to hard negatives or fusion.*
>
> We agree that hard negatives are crucial for learning TGIT, and we do not claim that FuseLIP would solve these tasks without them. Table 6 shows that hard negatives provide an important training signal for several tasks. However, **this does not explain the relative advantage of FuseLIP over late fusion, since the same hard-negative strategy is used for all models.**

---

> > ### Author Response · Authors · 2026-06-22
> > **Rebuttal [2/2]**
> >
> > > *the authors should include an analysis tying tokenizer fidelity and token budget to downstream accuracy and its implications on the inference cost.*
> >
> > We analyze the downstream effect of image-tokenizer fidelity in Table 13. Since both TiTok-S and TiTok-B use a fixed budget of 128 image tokens, this ablation isolates tokenizer fidelity from the number of image tokens. We observe that scaling from the S-sized to the B-sized tokenizer yields significant improvements, with further gains obtained from scaling the FuseLIP encoder. At the same time, our experiments do not vary the number of image tokens, so we do not claim a scaling law with respect to token budget. The token budget directly affects inference cost: using more image tokens would increase the sequence length processed by the encoder. Thus, improving tokenizer fidelity at a fixed token budget is particularly attractive, as it can improve downstream accuracy without increasing the encoder sequence length. We have included this discussion in Appendix B in the revised manuscript.
> >
> >  &nbsp;
> >
> > > *Please disambiguate the two uses of "masking" … and state explicitly that the encoder is bidirectional …*
> >
> > We have disambiguated the two uses of masking and the bidirectional nature of FuseLIP in the revised manuscript.
> >
> >  &nbsp;
> >
> > > *The contrastive loss is computed on the same masked input as the MMM loss. The authors should consider including an ablation that instead computes it on the unmasked input …*
> >
> > Our choice to compute both losses on the same masked input is motivated by efficiency: it allows us to combine contrastive learning and masked multimodal modeling in a single forward pass, without evaluating the encoder separately on masked and unmasked versions of the same sample. Our current MMM ablation in Table 6 shows that adding the masked modeling objective in this training setup **improves performance across almost all tasks as compared to using only the contrastive loss on unmasked input**, supporting our use of the shared masked input.
> >
> >  &nbsp;
> >
> > We thank the reviewer again for their constructive comments. We are happy to answer any further questions.